# A Damped Newton Method Achieves Global $\mathcal{O}\left(\frac{1}{k^2}\right)$ and Local Quadratic Convergence Rate

**Slavomír Hanzely**[*]
MBZUAI,[†] KAUST[‡]
shanzely@gmail.com

**Dmitry Kamzolov**[*]
MBZUAI[†]
kamzolov.opt@gmail.com

**Dmitry Pasechnyuk**
MBZUAI,[†] MIPT,[§] ISP RAS[¶]
dmivilensky1@gmail

**Alexander Gasnikov**
MIPT,[§] ISP RAS,[¶] HSE[‖]
gasnikov@yandex.ru

**Peter Richtárik**
KAUST[‡]
richtarik@gmail.com

**Martin Takáč**
MBZUAI[†]
takac.MT@gmail.com

## Abstract

In this paper, we present the first stepsize schedule for Newton method resulting in fast global and local convergence guarantees. In particular, a) we prove an $\mathcal{O}\left(\frac{1}{k^2}\right)$ global rate, which matches the state-of-the-art global rate of cubically regularized Newton method of Polyak and Nesterov (2006) and of regularized Newton method of Mishchenko (2021) and Doikov and Nesterov (2021), b) we prove a local quadratic rate, which matches the best-known local rate of second-order methods, and c) our stepsize formula is simple, explicit, and does not require solving any subproblem. Our convergence proofs hold under affine-invariance assumptions closely related to the notion of self-concordance. Finally, our method has competitive performance when compared to existing baselines, which share the same fast global convergence guarantees.

## 1 Introduction

Second-order optimization methods are the backbone of much of industrial and scientific computing. With origins that can be tracked back several centuries to the pioneering works of Newton [Newton, 1687], Raphson [Raphson, 1697] and Simpson [Simpson, 1740], they were extensively studied, generalized, modified, and improved in the last century [Kantorovich, 1948, Moré, 1978, Griewank, 1981a]. For a review of the historical development of the classical Newton-Raphson method, we refer the reader to the work of Ypma [1995]. The number of extensions and applications of second-order optimization methods is enormous; for example, the survey of Conn et al. [2000] on trust-region and quasi-Newton methods cited over a thousand papers.

### 1.1 Second-order methods and modern machine learning

Despite the rich history of the field, research on second-order methods has been flourishing up to this day. Some of the most recent development in the area was motivated by the needs of modern machine learning. Data-oriented machine learning depends on large datasets (both in number of features and number of datapoints), which are often stored in distributed/decentalized fashion. Consequently, there is a need for scalable algorithms.

---

[*]Equal contributions

[†]Mohamed bin Zayed University of Artificial Intelligence, United Arab Emirates

[‡]King Abdullah University of Science and Technology, Thuwal, Saudi Arabia

[§]Moscow Institute of Physics and Technology, Russia

[¶]ISP RAS Research Center for Trusted Artificial Intelligence, Russia

[‖]National Research University Higher School of Economics, Russia

36th Conference on Neural Information Processing Systems (NeurIPS 2022).

To tackle large number of features, Qu et al. [2016], Gower et al. [2019], Doikov and Richtárik [2018] and Hanzely et al. [2020] proposed variants of Newton method operating in random low-dimensional subspaces. On the other hand, Pilanci and Wainwright [2017], Xu et al. [2020] and Kovalev et al. [2019] developed subsampled Newton methods for solving empirical risk minimization (ERM) problems with large training datasets. Additionally, Bordes et al. [2009], Mokhtari and Ribeiro [2015], Gower et al. [2016], Byrd et al. [2016] and Kovalev et al. [2020] proposed stochastic variants of quasi-Newton methods. To tackle non-centralized nature of datasets, Shamir et al. [2014], Reddi et al. [2016], Wang et al. [2018] and Crane and Roosta [2019] considered distributed variants of Newton method, with improvements under various data/function similarity assumptions. Islamov et al. [2021], Safaryan et al. [2022], Qian et al. [2022], Islamov et al. [2022] and Agafonov et al. [2022] developed communication-efficient distributed variants of Newton method using the idea of communication compression and error compensation, without the need for any similarity assumptions.

We highlight two main research directions throughout of history of second-order methods: globally convergent methods under additional second-order smoothness (4) [Nesterov and Polyak, 2006] and local methods for self-concordant problems (12) [Nesterov and Nemirovski, 1989]. Former approach lead to various improvements such as acceleration Nesterov [2008], Monteiro and Svaiter [2013], usage of inexact information Ghadimi et al. [2017], Agafonov et al. [2020], generalization to tensor methods and their acceleration Nesterov [2021a], Gasnikov et al. [2019], Kovalev and Gasnikov [2022], superfast second-order methods under higher smoothness Nesterov [2021c,b], Kamzolov and Gasnikov [2020]. Latter approach was a breakthrough in 1990s, it lead to interior-point methods. Summary of the results can be found in books Nesterov and Nemirovski [1994], Nesterov [2018]. This direction is still popular up to this day Dvurechensky and Nesterov [2018], Hildebrand [2020], Doikov and Nesterov [2022b], Nesterov [2022].

As easy-to-scale alternative to second-order methods, first-order algorithms attracted a lot of attention. Many of their aspects have been explored, including strong results in variance reduction (Roux et al. [2012],Gower et al. [2020], Johnson and Zhang [2013], Nguyen et al. [2021, 2017]), preconditioning Jahani et al. [2022], acceleration (Nesterov [2013], d'Aspremont et al. [2021]) distributed/federated computation (Konečný et al. [2016], Chen et al. [2022], Berahas et al. [2016], Takáč et al. [2015], Richtárik and Takáč [2016], Kairouz et al. [2021]), and decentralized computation [Koloskova et al., 2020, Sadiev et al., 2021, Borodich et al., 2021]. However, the convergence of first-order methods always depends on the conditioning of the underlying problem. Improving conditioning is fundamentally impossible without using higher-order information. Removing this conditioning dependence is possible by incorporating information about the Hessian. This results in second-order methods. Their most compelling advantage is that they can converge extremely quickly, usually in just a few iterations.

## 1.2 Newton method: benefits and limitations

One of the most famous algorithms in optimization, Newton method, takes iterates of form

$$x_{k+1} = x_k - \left[\nabla^2 f(x_k)\right]^{-1} \nabla f(x_k). \tag{1}$$

Its iterates satisfy the recursion $\|\nabla f(x_{k+1})\|_2 \le c \|\nabla f(x_k)\|_2^2$ (for a constant $c > 0$), which means that Newton method converges locally quadratically. However, convergence of Newton method is limited to only to the neighborhood of the solution. It is well-known that when initialized far from optimum, Newton can diverge, both in theory and practice (Jarre and Toint [2016], Mascarenhas [2007]). We can explain intuition why this happens. Update rule of Newton (1) was chosen to minimize right hand side of Taylor approximation

$$f(y) \approx Q_f(y; x) \stackrel{\text{def}}{=} f(x) + \langle \nabla f(x), y - x \rangle + \frac{1}{2} \langle \nabla^2 f(x)(y - x), y - x \rangle. \tag{2}$$

The main problem is that Taylor approximation is not an upper bound, and therefore, global convergence of Newton method is not guaranteed.

## 1.3 Towards a fast globally convergent Newton method

Even though second-order algorithms with superlinear local convergence rates are very common, global convergence guarantees of any form are surprisingly rare. Many papers proposed globalization strategies, essentially all of them require some combination of the following: line-search, trust regions, damping/truncation, regularization. Some popular globalization strategies show non-increase

of functional value during the training. However, this turned out to be insufficient for convergence to the optimum. Jarre and Toint [2016], Mascarenhas [2007] designed simple functions (strictly convex with compact level sets) so that Newton method with Armijo stepsizes does not converge to the optimum. To this day, virtually all known global convergence guarantees are for regularized Newton methods with, which can be written as

$$x_{k+1} = x_k - \alpha_k \left( \nabla^2 f(x_k) + \lambda_k \mathbf{I} \right)^{-1} \nabla f(x_k), \tag{3}$$

where $\lambda_k \geq 0$. Parameter $\lambda_k$ is also known as Levenberg-Marquardt regularization [Moré, 1978], which was first introduced for a nonlinear least-squares objective. For simplicity, we disregard differences in the objectives for the literature comparison. Motivation behind update (3) is to replace Taylor approximation in (2) by an upper bound. The first method with proven global convergence rate $\mathcal{O}\left(k^{-2}\right)$ is Cubic Newton method [Nesterov and Polyak, 2006] for function $f$ with Lipschitz-continuous Hessian,

$$\|\nabla^2 f(x) - \nabla^2 f(y)\|_2 \leq L_2 \|x - y\|_2. \tag{4}$$

Under this condition, one can upper bound of Taylor approximation eq. (2) as

$$f(y) \leq Q_f(y; x) + \tfrac{L_2}{6} \|y - x\|_2^3. \tag{5}$$

Next iterate of Cubic Newton can be written as a minimizer of right hand side of (5)[7]

$$x_{k+1} = \underset{y \in \mathbb{E}}{\operatorname{argmin}} \left\{ Q_f(y; x_k) + \tfrac{L_2}{6} \|y - x_k\|_2^3 \right\}. \tag{6}$$

For our newly-proposed algorithm AICN (Algorithm 1), we are using almost identical step[7] [8]

$$x_{k+1} = \underset{y \in \mathbb{E}}{\operatorname{argmin}} \left\{ Q_f(y; x_k) + \tfrac{L_{\text{semi}}}{6} \|y - x_k\|_{x_k}^3 \right\}. \tag{7}$$

The difference between the update of Cubic Newton and AICN is that we measure the cubic regularization term in the local Hessian norms. This seemingly negligible perturbation turned out to be of a great significance for two reasons **a)** model in (7) is affine-invariant, **b)** surprisingly, the next iterate of (7) lies in the direction of Newton method step and is obtainable without regularizer $\lambda_k$ (AICN just needs to set stepsize $\alpha_k$). We elaborate on both of these points later in the paper.

Cubic Newton method (6) can be equivalently expressed in form (3) with $\alpha_k = 1$ and $\lambda_k = L_2 \|x_k - x_{k+1}\|_2$. However, since such $\lambda_k$ depends on $x_{k+1}$, resulting algorithm requires additional subroutine for solving its subproblem each iteration. Next work showing convergence rate of regularized Newton method [Polyak, 2009] avoided implicit steps by choosing $\lambda_k \propto \|\nabla f(x_k)\|_2$. However, this came with a trade-off for slower convergence rate, $\mathcal{O}\left(k^{-1/4}\right)$. Finally, Mishchenko [4/2021] (see also the work of Doikov and Nesterov [12/2021]) improved upon both of these works by using explicit regularization $\alpha_k = 1$, $\lambda_k \propto \sqrt{L_2 \|\nabla f(x_k)\|_2}$, and proving global rate $\mathcal{O}\left(k^{-2}\right)$.

## 2 Contributions

### 2.1 AICN as a damped Newton method

In this work, we investigate global convergence for most basic globalization strategy, stepsized Newton method without any regularizer ($\lambda_k = 0$). This algorithm is also referred as damped (or truncated) Newton method; it can be written as

$$x_{k+1} = x_k - \alpha_k \nabla^2 f(x_k)^{-1} \nabla f(x_k).$$

Resulting algorithm was investigated in detail as an interior-point method. Nesterov [2018] shows quadratic local convergence for stepsizes $\alpha_1 \stackrel{\text{def}}{=} \frac{1}{1+G_1}, \alpha_2 \stackrel{\text{def}}{=} \frac{1+G_1}{1+G_1+G_1^2}$, where[9][10]

---

[7]Where $\mathbb{E}$ a $d$-dimensional Euclidean space, defined in Section 2.3.

[8]Function $f$ is $L_{\text{semi}}$-semi-strongly self-concordant (Definition 3). Instead of $L_{\text{semi}}$, we will use its upper bound $L_{\text{est}}$, $L_{\text{est}} \geq L_{\text{semi}}$.

[9]Function $f$ is $L_{\text{sc}}$-self-concordant (Definition 1).

[10]Dual norm $\|\nabla f(x_k)\|_{x_k}^* = \left\langle \nabla f(x_k), \nabla^2 f(x_k)^{-1} \nabla f(x_k) \right\rangle$ is defined in Section 2.3.

Table 1: A summary of regularized Newton methods with global convergence guarantees. We consider algorithms with updates of form $x_{k+1} = x_k - \alpha_k \left(\nabla^2 f(x_k) + \lambda_k \mathbf{I}\right)^{-1} \nabla f(x_k)$. For simplicity of comparison, we disregard differences in objectives and assumptions. We assume $L_2$-smoothness of Hessian, $L_{\text{semi}}$-semi-strong self-concordance, convexity (Definition 3), $\mu$-strong convexity locally and bounded level sets. For regularization parameter holds $\lambda_k \geq 0$ and stepsize satisfy $0 < \alpha_k \leq 1$. We highlight the best know rates in blue.

| Algorithm | Regularizer $\lambda_k \propto$ | Stepsize $\alpha_k =$ | Affine[1] invariant? (alg., ass., rate) | Avoids line search? | Global convergence rate | Local[1] convergence exponent | Reference |
|---|---|---|---|---|---|---|---|
| Newton | 0 | 1 | ($\checkmark$, $\times$, $\times$) | $\checkmark$ | $\times$ | 2 | Kantorovich [1948] |
| Newton | 0 | 1 | ($\checkmark$, $\checkmark$, $\checkmark$) | $\checkmark$ | $\times$ | 2 | Nesterov and Nemirovski [1994] |
| Damped Newton B | 0 | $\frac{1}{1+G_1}$[4] | ($\checkmark$, $\checkmark$, $\checkmark$) | $\checkmark$ | $\mathcal{O}\left(k^{-\frac{1}{2}}\right)$ | 2 | Nesterov [2018, (5.1.28)] |
| Damped Newton C | 0 | $\frac{1+G_1}{1+G_1+G_1^2}$[4] | ($\checkmark$, $\checkmark$, $\checkmark$) | $\checkmark$ | $\times$ | 2 | Nesterov [2018, (5.2.1)$_C$] |
| Cubic Newton | $L_2\|x_{k+1} - x_k\|_2$ | 1 | ($\times$, $\times$, $\times$) | $\times$ | $\mathcal{O}\left(k^{-2}\right)$ | 2 | Nesterov and Polyak [2006], Griewank [1981b], Doikov and Nesterov [2022a] |
| Locally Reg. Newton | $\|\nabla f(x_k)\|_2$ | 1 | ($\times$, $\times$, $\times$) | $\checkmark$ | $\times$ | 2 | Polyak [2009] |
| Globally Reg. Newton | $\|\nabla f(x_k)\|_2$ | $\frac{\mu+\|\nabla f(x_k)\|_2}{L_1}$[3] | ($\times$, $\times$, $\times$) | $\times$ | $\mathcal{O}\left(k^{-\frac{1}{4}}\right)$ | 2 | Polyak [2009] |
| Globally Reg. Newton | $\sqrt{L_2\|\nabla f(x_k)\|_2}$ | 1 | ($\times$, $\times$, $\times$) | $\checkmark$ | $\mathcal{O}\left(k^{-2}\right)$ | $\frac{3}{2}$ | Mishchenko [4/2021] Doikov and Nesterov [12/2021] |
| **AIC Newton** (Algorithm 1) | 0 | $\frac{-1+\sqrt{1+2G}}{G}$[4] | ($\checkmark$, $\checkmark$, $\checkmark$) | $\checkmark$ | $\mathcal{O}\left(k^{-2}\right)$ | 2 | **This work** |

[1] In triplets, we report whether algorithm, used assumptions, convergence rate are affine-invariant, respectively.

[2] For a Lyapunov function $\Phi^k$ and a constant $c$, we report exponent $\beta$ of $\Phi(x_{k+1}) \leq c\Phi(x_k)^\beta$.

[3] $f$ has $L_1$-Lipschitz continuous gradient.

[4] For simplicity, we denote $G_1 \stackrel{\text{def}}{=} L_{\text{sc}}\|\nabla f(x_k)\|_{x_k}^*$ and $G \stackrel{\text{def}}{=} L_{\text{semi}}\|\nabla f(x_k)\|_{x_k}^*$ (for $L_{\text{est}} \leftarrow L_{\text{semi}}$).

$G_1 \stackrel{\text{def}}{=} L_{\text{sc}}\|\nabla f(x_k)\|_{x_k}^*$. Our algorithm AICN is also damped Newton method with stepsize $\alpha = \frac{-1+\sqrt{1+2G}}{G}$, where[810] $G \stackrel{\text{def}}{=} L_{\text{semi}}\|\nabla f(x_k)\|_{x_k}^*$. Mentioned stepsizes $\alpha_1, \alpha_2, \alpha$ share two characteristics. Firstly, all of them depends on gradient computed in the dual norm and scaled by a smoothness constant ($G_1$ or $G$). Secondly, all of these stepsizes converge to 1 from below (for $\hat{\alpha} \in \{\alpha_1, \alpha_2, \alpha\}$ holds $0 < \hat{\alpha} \leq 1$ and $\lim_{x \to x_*} \hat{\alpha} = 1$). Our algorithm uses stepsize bigger by orders of magnitude (see Figure 3 in Appendix A for detailed comparison). The main difference between already established stepsizes $\alpha_1, \alpha_2$ and our stepsize $\alpha$ are resulting global convergence rates. While stepsize $\alpha_2$ does not lead to a global convergence rate, and $\alpha_1$ leads to rate $\mathcal{O}\left(k^{-1/2}\right)$, our stepsize $\alpha$ leads to a significantly faster, $\mathcal{O}\left(k^{-2}\right)$ rate. Our rate matches best known global rates for regularized Newton methods. We manage to achieve these results by carefully choosing assumptions. While rates for $\alpha_1$ and $\alpha_2$ follows from standard self-concordance, our assumptions are a consequence of a slightly stronger version of self-concordance. We will discuss this difference in detail later.

We summarize important properties of regularized Newton methods with fast global convergence guarantees and damped Newton methods in Table 1.

## 2.2 Summary of contributions

To summarize novelty in our work, we present a novel algorithm AICN. Our algorithm can be interpreted in two viewpoints **a)** as a regularized Newton method (version of Cubic Newton method), **b)** as a damped Newton method. AICN enjoys the best properties of these two worlds:

- **Fast global convergence**: AICN converges globally with rate $\mathcal{O}\left(k^{-2}\right)$ (Theorem 2, 4), which matches state-of-the-art global rate for all regularized Newton methods. Furthermore, it is the first such rate for Damped Newton method.

- **Fast local convergence:** In addition to the fast global rate, AICN decreases gradient norms locally in quadratic rate (Theorem 3). This result matches the best-known rates for both regularized Newton algorithms and damped Newton algorithms.

- **Simplicity:** Previous works on Newton regularizations can be viewed as a popular global-convergence fix for the Newton method. We propose an even simpler fix in the form of a stepsize schedule (Section 3).

- **Implementability:** Step of AICN depends on a smoothness constant $L_{\text{semi}}$ (Definition 3). Given this constant, next iterate of AICN can be computed directly.

  This is improvement over Cubic Newton [Nesterov and Polyak, 2006], which for a given constant $L_2$ needs to run **line-search** subroutine each iteration to solve its subproblem

- **Improvement:** Avoiding latter subroutine yields theoretical improvements. If we compute matrix inverses naively, iteration cost of AICN is $\mathcal{O}(d^3)$ (where $d$ is a dimension of the problem), which is improvement over $\mathcal{O}(d^3 \log \varepsilon^{-1})$ iteration cost of Cubic Newton [Nesterov and Polyak, 2006].

- **Practical performance:** We show that in practice, AICN outperforms all algorithms sharing same convergence guarantees: Cubic Newton [Nesterov and Polyak, 2006] and Globally Regularized Newton [Mishchenko, 4/2021] and Doikov and Nesterov [12/2021], and fixed stepsize Damped Newton method (Section 5).

- **Geometric properties:** We analyze AICN under more geometrically natural assumptions. Instead of smoothness, we use a version of self-concordance (Section 3.1), which is invariant to affine transformations and hence also to a choice of a basis. AICN preserves affine-invariance obtained from assumptions throughout the convergence. In contrast, Cubic Newton uses base-dependent $l_2$ norm and hence depends on a choice of a basis. This represents an extra layer of complexity.

- **Alternative analysis:** We also provide alternative analysis under weaker assumptions (Appendix C).

The rest of the paper is structured as follows. In Section 2.3 we introduce our notation. In Section 3, we discuss algorithm AICN, affine-invariant properties and self-concordance. In Sections 4.1 and 4.2 we show global and local convergence guarantees, respectively. In Section 5 we present an empirical comparison of AICN with other algorithms sharing fast global convergence.

## 2.3 Minimization problem & notation

> In the paper, we consider a $d$-dimensional Euclidean space $\mathbb{E}$. Its dual space, $\mathbb{E}^*$, is composed of all linear functionals on $\mathbb{E}$. For a functional $g \in \mathbb{E}^*$, we denote by $\langle g, x \rangle$ its value at $x \in \mathbb{E}$.

We consider the following convex optimization problem:

$$\min_{x \in \mathbb{E}} f(x), \tag{8}$$

where $f(x) \in C^2$ is a convex function with continuous first and second derivatives and positive definite Hessian. We assume that the problem has a unique minimizer $x_* \in \operatorname*{argmin}_{x \in \mathbb{E}} f(x)$. Note, that $\nabla f(x) \in \mathbb{E}^*$, $\nabla^2 f(x)h \in \mathbb{E}^*$. Now, we introduce different norms for spaces $\mathbb{E}$ and $\mathbb{E}^*$. Denote $x, h \in \mathbb{E}, g \in \mathbb{E}^*$. For a self-adjoint positive-definite operator $\mathbf{H} : \mathbb{E} \to \mathbb{E}^*$, we can endow these spaces with conjugate Euclidean norms:

$$\|x\|_{\mathbf{H}} \overset{\text{def}}{=} \langle \mathbf{H}x, x \rangle^{1/2}, \, x \in \mathbb{E}, \qquad \|g\|_{\mathbf{H}}^* \overset{\text{def}}{=} \langle g, \mathbf{H}^{-1}g \rangle^{1/2}, \, g \in \mathbb{E}^*.$$

For identity $\mathbf{H} = \mathbf{I}$, we get classical Euclidean norm $\|x\|_{\mathbf{I}} = \langle x, x \rangle^{1/2}$. For local Hessian norm $\mathbf{H} = \nabla^2 f(x)$, we use shortened notation

$$\|h\|_x \overset{\text{def}}{=} \langle \nabla^2 f(x)h, h \rangle^{1/2}, \, h \in \mathbb{E}, \qquad \|g\|_x^* \overset{\text{def}}{=} \langle g, \nabla^2 f(x)^{-1}g \rangle^{1/2}, \, g \in \mathbb{E}^*. \tag{9}$$

Operator norm is defined by

$$\|\mathbf{H}\|_{op} \overset{\text{def}}{=} \sup_{v \in \mathbb{E}} \frac{\|\mathbf{H}v\|_x^*}{\|v\|_x}, \tag{10}$$

for $\mathbf{H} : \mathbb{E} \to \mathbb{E}^*$ and a fixed $x \in \mathbb{E}$. If we consider a specific case $\mathbb{E} \leftarrow \mathbb{R}^d$, then $\mathbf{H}$ is a symmetric positive definite matrix.

# 3 New Algorithm: Affine-Invariant Cubic Newton

Finally, we are ready to present algorithm AICN. It is damped Newton method with updates

$$x_{k+1} = x_k - \alpha_k \nabla^2 f(x_k)^{-1} \nabla f(x_k), \tag{11}$$

with stepsize

$$\alpha_k \stackrel{\text{def}}{=} \frac{-1 + \sqrt{1 + 2L_{\text{est}} \|\nabla f(x_k)\|_{x_k}^*}}{L_{\text{est}} \|\nabla f(x_k)\|_{x_k}^*},$$

as summarized in Algorithm 1. Stepsize satisfy $\alpha_k \leq 1$ (from AG inequality, (45)). Also $\lim_{x_k \to x_*} \alpha_k = 1$, hence (11) converges to Newton method. Next, we are going to discuss geometric properties of our algorithm.

---

**Algorithm 1** AICN: Affine-Invariant Cubic Newton

1: **Requires:** Initial point $x_0 \in \mathbb{E}$, constant $L_{\text{est}}$ s.t. $L_{\text{est}} \geq L_{\text{semi}} > 0$
2: **for** $k = 0, 1, 2 \ldots$ **do**
3: $\quad \alpha_k = \frac{-1 + \sqrt{1 + 2L_{\text{est}} \|\nabla f(x_k)\|_{x_k}^*}}{L_{\text{est}} \|\nabla f(x_k)\|_{x_k}^*}$
4: $\quad x_{k+1} = x_k - \alpha_k \left[ \nabla^2 f(x_k) \right]^{-1} \nabla f(x_k)$ $\qquad\qquad$ ▷ Note that $x_{k+1} \stackrel{(17)}{=} S_{f, L_{\text{est}}}(x_k)$.
5: **end for**

---

## 3.1 Geometric properties: affine invariance

One of the main geometric properties of the Newton method is *affine invariance*, invariance to affine transformations of variables. Let $\mathbf{A} : \mathbb{E} \to \mathbb{E}^*$ be a non-degenerate linear transformation. Consider function $\phi(y) = f(\mathbf{A}y)$. By affine transformation, we denote $f(x) \to \phi(y) = f(\mathbf{A}y), x \to \mathbf{A}^{-1}y$.

**Significance of norms:** Note that local Hessian norm $\|h\|_{\nabla f(x)}$ is affine-invariant because

$$\|z\|_{\nabla^2 \phi(y)}^2 = \left\langle \nabla^2 \phi(y)z, z \right\rangle = \left\langle \mathbf{A}^\top \nabla^2 f(\mathbf{A}y) \mathbf{A} z, z \right\rangle = \left\langle \nabla^2 f(x) h, h \right\rangle = \|h\|_{\nabla^2 f(x)}^2,$$

where $h = \mathbf{A}z$. On the other hand, induced norm $\|h\|_{\mathbf{I}}$ is not affine-invariant because

$$\|z\|_{\mathbf{I}}^2 = \langle z, z \rangle = \left\langle \mathbf{A}^{-1}h, \mathbf{A}^{-1}h \right\rangle = \|\mathbf{A}^{-1}h\|_{\mathbf{I}}^2.$$

With respect to geometry, the most natural norm is local Hessian norm, $\|h\|_{\nabla f(x)}$. From affine invariance follows that for this norm, the level sets $\left\{ y \in \mathbb{E} \mid \|y - x\|_x^2 \leq c \right\}$ are balls centered around $x$ (all directions have the same scaling). In comparison, scaling of the $l_2$ norm is dependent on eigenvalues of the Hessian. In terms of convergence, one direction in $l_2$ can significantly dominate others and slow down an algorithm.

**Significance for algorithms:** Algorithms that are not affine-invariant can suffer from chosen coordinate system. This is the case for Cubic Newton, as its model (4) is bound to base-dependent $l_2$ norm. Same is true for any other method regularized with an induced norm $\|h\|_{\mathbf{I}}$. On the other hand, (damped) Newton methods have affine-invariant models, and hence as algorithms independent of the chosen coordinate system. We prove this claim in following lemma (note: $\alpha_k = 1$ and $\alpha_k$ from (11) are affine-invariant).

**Lemma 1** (Lemma 5.1.1 Nesterov [2018]). *Let the sequence $\{x_k\}$ be generated by a damped Newton method with affine-invariant stepsize $\alpha_k$, applied to the function $f$: $x_{k+1} = x_k - \alpha_k \left[ \nabla^2 f(x_k) \right]^{-1} \nabla f(x_k)$. For function $\phi(y)$, damped Newton method generates $\{y_k\}$: $y_{k+1} = y_k - \alpha_k \left[ \nabla^2 \phi(y_k) \right]^{-1} \nabla \phi(y_k)$, with $y_0 = \mathbf{A}^{-1}x_0$. Then $y_k = \mathbf{A}^{-1}x_k$.*

## 3.2 Significance in assumptions: self-concordance

We showed that damped Newton methods preserve affine-invariance through iterations. Hence it is more fitting to analyze them under affine-invariant assumptions. Affine-invariant version of smoothness, *self-concordance*, was introduced in Nesterov and Nemirovski [1994].

**Definition 1.** *Convex function $f \in C^3$ is called self-concordant if*

$$|D^3 f(x)[h]^3| \leq L_{sc}\|h\|_x^3, \quad \forall x, h \in \mathbb{E}, \tag{12}$$

*where for any integer $p \geq 1$, by $D^p f(x)[h]^p \stackrel{def}{=} D^p f(x)[h, \ldots, h]$ we denote the p-th order directional derivative[a] of $f$ at $x \in \mathbb{E}$ along direction $h \in \mathbb{E}$.*

---
[a]For example, $D^1 f(x)[h] = \langle \nabla f(x), h \rangle$ and $D^2 f(x)[h]^2 = \langle \nabla^2 f(x)h, h \rangle$.

Both sides of inequality are affine-invariant. This assumption corresponds to a big class of optimization methods called interior-point methods. Self-concordance implies uniqueness of the solution, as stated in following proposition.

**Proposition 1** (Theorem 5.1.16, Nesterov [2018]). *Let a self-concordant function $f$ be bounded below. Then it attains its minimum at a single point.*

Rodomanov and Nesterov [2021] introduced stronger version of self-concordance assumption.

**Definition 2.** *Convex function $f \in C^2$ is called strongly self-concordant if*

$$\nabla^2 f(y) - \nabla^2 f(x) \preceq L_{str}\|y - x\|_z \nabla^2 f(w), \quad \forall y, x, z, w \in \mathbb{E}. \tag{13}$$

For our analysis, we introduce definition for functions between self-concordant and strongly self-concordant.

**Definition 3.** *Convex function $f \in C^2$ is called semi-strongly self-concordant if*

$$\left\|\nabla^2 f(y) - \nabla^2 f(x)\right\|_{op} \leq L_{semi}\|y - x\|_x, \quad \forall y, x \in \mathbb{E}. \tag{14}$$

Note that all of the Definitions 1 - 3 are affine-invariant, their respective classes satisfy

*strong self-concordance $\subseteq$ semi-strong self-concordance $\subseteq$ self-concordance.*

Also, for a fixed strongly self-concordant function $f$ and smallest such $L_{sc}, L_{semi}, L_{str}$ holds $L_{sc} \leq L_{semi} \leq L_{str}$.

These notions are related to the convexity and smoothness; strong concordance follows from function $L_2$-Lipschitz continuous Hessian and strong convexity.

**Proposition 2** (Example 4.1 from [Rodomanov and Nesterov, 2021]). *Let $\mathbf{H} : \mathbb{E} \to \mathbb{E}^*$ be a self-adjoint positive definite operator. Suppose there exist $\mu > 0$ and $L_2 \geq 0$ such that the function $f$ is $\mu$-strongly convex and its Hessian is $L_2$-Lipschitz continuous (4) with respect to the norm $\|\cdot\|_{\mathbf{H}}$. Then $f$ is strongly self-concordant with constant $L_{str} = \frac{L_2}{\mu^{3/2}}$.*

### 3.3 From assumptions to algorithm

From semi-strong self-concordance we can get a second-order bounds on the function and model.

**Lemma 2.** *If $f$ is semi-strongly self-concordant, then*

$$|f(y) - Q_f(y; x)| \leq \tfrac{L_{semi}}{6}\|y - x\|_x^3, \quad \forall x, y \in \mathbb{E}. \tag{15}$$

*Consequently, we have upper bound for function value in form*

$$f(y) \leq Q_f(y; x) + \tfrac{L_{semi}}{6}\|y - x\|_x^3. \tag{16}$$

One can show that (16) is not valid for just self-concordant functions. For example, there is no such upper bound for $-\log(x)$. Hence, the semi-strongly self-concordance is significant as an assumption.

We can define iterates of optimization algorithm to be minimizers of the right hand side of (16),

$$S_{f,L_{\text{est}}}(x) \stackrel{\text{def}}{=} x + \underset{h \in \mathbb{E}}{\arg\min} \left\{ f(x) + \langle \nabla f(x), h \rangle + \tfrac{1}{2} \langle \nabla^2 f(x) h, h \rangle + \tfrac{L_{\text{est}}}{6} \|h\|_x^3 \right\}, \qquad (17)$$

$$x_{k+1} = S_{f,L_{\text{est}}}(x_k), \qquad (18)$$

for an estimate constant $L_{\text{est}} \geq L_{\text{semi}}$. It turns out that subproblem (17) is easy to solve. To get an explicit solution, we compute its gradient w.r.t. $h$. For solution $h^*$, it should be equal to zero,

$$\nabla f(x) + \nabla^2 f(x) h^* + \tfrac{L_{\text{est}}}{2} \|h^*\|_x \nabla^2 f(x) h^* = 0, \qquad (19)$$

$$h^* = - \left[ \nabla^2 f(x) \right]^{-1} \nabla f(x) \cdot \left( \tfrac{L_{\text{est}}}{2} \|h^*\|_x + 1 \right)^{-1}. \qquad (20)$$

We get that step (18) has the same direction as a Newton method and is scaled by $\alpha_k = \left( \tfrac{L_{\text{est}}}{2} \|h^*\|_x + 1 \right)^{-1}$. Now, we substitute $h^*$ from (20) to (19)

$$\nabla f(x) - \nabla f(x) \alpha_k - \tfrac{L_{\text{est}}}{2} \left\langle \nabla f(x), \left[ \nabla^2 f(x) \right]^{-1} \nabla f(x) \right\rangle^{1/2} \nabla f(x) \alpha_k^2 = 0,$$

$$\nabla f(x) \left( 1 - \alpha_k - \tfrac{L_{\text{est}}}{2} \|\nabla f(x)\|_x^* \alpha_k^2 \right) = 0. \qquad (21)$$

We solve the quadratic equation (21) for $\alpha_k$, and obtain explicit formula for stepsizes of AICN, as (11). We formalize this connection in theorem, for further explanation see proof in Appendix B.

---

**Theorem 1.** *For $L_{est} \geq L_{semi}$, update of* AICN *(11),*

$$x_{k+1} = x_k - \alpha_k \nabla^2 f(x_k)^{-1} \nabla f(x_k), \qquad \text{where} \qquad \alpha_k = \frac{-1 + \sqrt{1 + 2L_{est} \|\nabla f(x_k)\|_{x_k}^*}}{L_{est} \|\nabla f(x_k)\|_{x_k}^*},$$

*is a minimizer of upper bound (17), $x_{k+1} = S_{f,L_{est}}(x_k)$.*

---

## 4 Convergence Results

### 4.1 Global convergence

Next, we focus on global convergence guarantees. We will utilize the following assumption:

---

**Assumption 1** (Bounded level sets)**.** *The objective function $f$ has a unique minimizer $x_*$. Also, the diameter of the level set $\mathcal{L}(x_0) \stackrel{\text{def}}{=} \{x \in \mathbb{E} : f(x) \leq f(x_0)\}$ is bounded by a constant $D_2$ as[a], $\underset{x \in \mathcal{L}(x_0)}{\max} \|x - x_*\|_2 \leq D_2 < +\infty$.*

---
[a]We state it in $l_2$ norm for easier verification. In proofs, we use its variant $D$ in Hessian norms, (23).

---

Our analysis proceeds as follows. Firstly, we show that one step of the algorithm decreases function value, and secondly, we use the technique from [Ghadimi et al., 2017] to show that multiple steps lead to $\mathcal{O}\left(k^{-2}\right)$ global convergence. We start with following lemma.

**Lemma 3** (One step globally)**.** *Let function $f$ be $L_{semi}$-semi-strongly self-concordant, convex with positive-definite Hessian and $L_{est} \geq L_{semi}$. Then for any $x \in \mathbb{E}$, we have*

$$S_{f,L_{est}}(x) \leq \underset{y \in \mathbb{E}}{\min} \left\{ f(y) + \tfrac{L_{est}}{3} \|y - x\|_x^3 \right\}. \qquad (22)$$

This lemma implies that step (17) decreases function value (take $y \leftarrow x$). Using notation of Assumption 1, $x_k \in \mathcal{L}(x_0)$ for any $k \geq 0$. Also, setting $y \leftarrow x$ and $x \leftarrow x_*$ in (14) yields $\|x - x_*\|_x \leq \left( \|x - x_*\|_{x_*}^2 + L_{\text{est}} \|x - x_*\|_{x_*}^3 \right)^{\frac{1}{2}}$. We denote those distances $D$ and $R$,

$$D \stackrel{\text{def}}{=} \underset{t \in [0; k+1]}{\max} \|x_t - x_*\|_{x_t} \qquad \text{and} \qquad R \stackrel{\text{def}}{=} \underset{x \in \mathcal{L}(x_0)}{\max} \left( \|x - x_*\|_{x_*}^2 + L_{\text{est}} \|x - x_*\|_{x_*}^3 \right)^{\frac{1}{2}}. \qquad (23)$$

They are both affine-invariant and $R$ upper bounds $D$. While $R$ depends only on the level set $\mathcal{L}(x_0)$, $D$ can be used to obtain more tight inequalities. We avoid using common distance $D_2$, as $l_2$ norm would ruin affine-invariant properties.

**Theorem 2.** *Let $f(x)$ be a $L_{semi}$-semi-strongly self-concordant convex function with positive-definite Hessian, constant $L_{est}$ satisfy $L_{est} \geq L_{semi}$ and Assumption 1 holds. Then, after $k + 1$ iterations of Algorithm 1, we have the following convergence rate:*

$$f(x_{k+1}) - f(x_*) \leq O\left(\frac{L_{est}D^3}{k^2}\right) \leq O\left(\frac{L_{est}R^3}{k^2}\right). \tag{24}$$

Consequently, AICN converges globally with a fast rate $\mathcal{O}\left(k^{-2}\right)$. We can now present local analysis.

### 4.2 Local convergence

For local quadratic convergence are going to utilise following lemmas.

**Lemma 4.** *For convex $L_{semi}$-semi-strongly self-concordant function $f$ and for any $0 < c < 1$ in the neighborhood of solution*

$$x_k \in \left\{x : \|\nabla f(x)\|_x^* \leq \frac{(2c+1)^2-1}{2L_{est}}\right\} \text{ holds } \nabla^2 f(x_{k+1})^{-1} \preceq (1-c)^{-2}\nabla^2 f(x_k)^{-1}. \tag{25}$$

Lemma 4 formalizes that a inverse hessians of a self-concordant function around the solution is non-degenerate. With this result, we can show one-step gradient norm decrease.

**Lemma 5** (One step decrease locally)**.** *Let function $f$ be $L_{semi}$-semi-strongly self-concordant and $L_{est} \geq L_{semi}$. If $x_k$ such that (25) holds, then for next iterate $x_{k+1}$ of AICN holds*

$$\|\nabla f(x_{k+1})\|_{x_k}^* \leq L_{est}\alpha_k^2\|\nabla f(x_k)\|_{x_k}^{*2} < L_{est}\|\nabla f(x_k)\|_{x_k}^{*2}. \tag{26}$$

*Using Lemma 4, we shift the gradient bound to respective norms,*

$$\|\nabla f(x_{k+1})\|_{x_{k+1}}^* \leq \frac{L_{est}\alpha_k^2}{1-c}\|\nabla f(x_k)\|_{x_k}^{*2} < \frac{L_{est}\alpha_k^2}{1-c}\|\nabla f(x_k)\|_{x_k}^{*2}. \tag{27}$$

*Gradient norm decreases $\|\nabla f(x_{k+1})\|_{x_{k+1}}^* \leq \|\nabla f(x_k)\|_{x_k}^*$ for $\|\nabla f(x_k)\|_{x_k}^* \leq \frac{(2-c)^2-1}{2L_{est}}$.*

As a result, neighbourhood of the local convergence is $\left\{x : \|\nabla f(x)\|_x^* \leq \min\left[\frac{(2-c)^2-1}{2L_{est}}; \frac{(2c+1)^2-1}{2L_{est}}\right]\right\}$. Maximizing by $c$, we get $c = 1/3$ and neighrbourhood $\left\{x : \|\nabla f(x)\|_x^* \leq \frac{8}{9L_{est}}\right\}$. One step of AICN decreases gradient norm quadratically, multiple steps leads to following decrease.

**Theorem 3** (Local convergence rate)**.** *Let function $f$ be $L_{semi}$-semi-strongly self-concordant, $L_{est} \geq L_{semi}$ and starting point $x_0$ be in the neighborhood of the solution such that $\|\nabla f(x_0)\|_{x_0}^* \leq \frac{8}{9L_{est}}$. For $k \geq 0$, we have quadratic decrease of the gradient norms,*

$$\|\nabla f(x_k)\|_{x_k}^* \leq \left(\frac{3}{2}L_{est}\right)^k \left(\|\nabla f(x_0)\|_{x_0}^*\right)^{2^k}. \tag{28}$$

## 5 Numerical Experiments

In this section, we evaluate proposed AICN (Algorithm 1) algorithm on the logistic regression task and second-order lower bound function. We compare it with regularized Newton methods sharing fast global convergence guarantees: Cubic Newton method [Nesterov and Polyak, 2006], and Globally Regularized Newton method ([Mishchenko, 4/2021, Doikov and Nesterov, 12/2021]) with $L_2$-constant. Because AICN has a form of a damped Newton method, we also compare it with Damped Newton with fixed (tuned) stepsize $\alpha_k = \alpha$. We report decrease in function value $f(x_k)$, function value suboptimality $f(x_k) - f(x_*)$ with respect to iteration and time. The methods are implemented as PyTorch optimizers. The code is available at https://github.com/OPTAMI.

### 5.1 Logistic regression

For first part, we solve the following empirical risk minimization problem:

$$\min_{x \in \mathbb{R}^d} \left\{f(x) = \frac{1}{m}\sum_{i=1}^m \log\left(1 - e^{-b_i a_i^\top x}\right) + \frac{\mu}{2}\|x\|_2^2\right\},$$

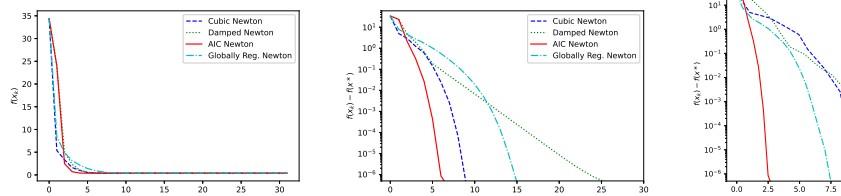

Figure 1: Comparison of regularized Newton methods and Damped Newton method for logistic regression task on *a9a* dataset.

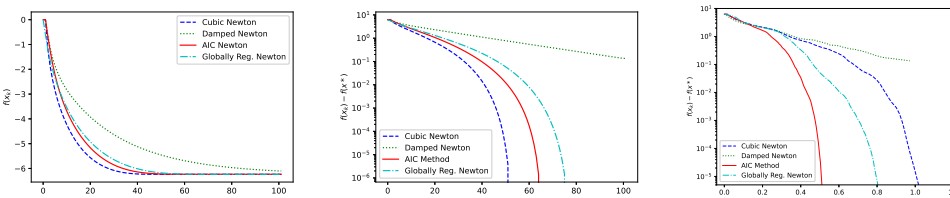

Figure 2: Comparison of regularized Newton methods and Damped Newton method for *second-order lower bound function*.

where $\{(a_i, b_i)\}_{i=1}^m$ are given data samples described by features $a_i$ and class $b_i \in \{-1, 1\}$.

In Figure 1, we consider task of classification images on dataset *a9a* [Chang and Lin, 2011]. Number of features for every data sample is $d = 123$, $m = 20000$. We take staring point $x_0 \stackrel{\text{def}}{=} 10[1, 1, \ldots, 1]^\top$ and $\mu = 10^{-3}$. Our choice is differ from $x_0 = 0$ (equal to all zeroes) to show globalisation properties of methods ($x_0 = 0$ is very close to the solution, as Newton method converges in 4 iterations). Parameters of all methods are fine-tuned, we choose parameters $L_{\text{est}}, L_2, \alpha$ (of AICN, Cubic Newton, Damped Newton, resp.) to largest values having monotone decrease in reported metrics. Fine-tuned values are $L_{\text{est}} = 0.97$ $L_2 = 0.000215$, $\alpha = 0.285$. Figure 1 demonstrates that AICN converges slightly faster than Cubic Newton method by iteration, notably faster than Globally Regularized Newton and significantly faster than Damped Newton. AICN outperforms every method by time.

## 5.2 Second-order lower bound function

For second part we solve the following minimization problem:

$$\min_{x \in \mathbb{R}^d} \left\{ f(x) = \frac{1}{d} \sum_{j=1}^d \left| [\mathbf{A}x]_j \right|^3 - x_1 + \frac{\mu}{2} \|x\|_2^2 \right\}, \quad \text{where} \quad \mathbf{A} = \begin{pmatrix} 1 & -1 & 0 & \ldots & 0 \\ 0 & 1 & -1 & \ldots & 0 \\ & & \ldots & \ldots & \\ 0 & 0 & \ldots & 0 & 1 \end{pmatrix}.$$

This function is a lower bound for a class of functions with Lipschitz continuous Hessian (4) with additional regularization [Nesterov and Polyak, 2006, Nesterov, 2021a]. In Figure 2, we take $d = 20$, $x_0 = 0$ (equal to all zeroes). Parameters $L_{\text{est}}, L_2, \alpha$ are fine-tuned to largest values having monotone decrease in reported metrics: $L_{\text{est}} = 662$ $L_2 = 0.662$, $\alpha = 0.0172$. Figure 2 demonstrates that AICN converges slightly slower than Cubic Newton method, slightly faster than Globally Regularized Newton, and significantly faster than Damped Newton. ain outperforms every method by time. More experiments are presented in Appendix A. Note, that the iteration of the Cubic Newton method needs an additional line-search, so one iteration of Cubic Newton is computationally harder than one iteration of AICN. More experiments are presented in Appendix A.

## Acknowledgement

The work of D. Pasechnyuk and A. Gasnikov was supported by a grant for research centers in the field of artificial intelligence, provided by the Analytical Center for the Government of the RF in accordance with the subsidy agreement (agreement identifier 000000D730321P5Q0002) and the agreement with the Ivannikov Institute for System Programming of the RAS dated November 2, 2021 No. 70-2021-00142.

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
