

Figure 3: Comparison of stepsizes of affine-invariant damped Newton methods with quadratic local convergence. We compare AICN (blue), and stepsizes from Nesterov [2018] in orange and green. We set $L_{\text{semi}} = L_{\text{sc}} = 5$.

# Appendix

## A    Extra comparisons to other methods

### A.1    Damped Newton method stepsize comparison

In Figure 3, we present comparison of stepizes of AICN with other damped Newton methods [Nesterov, 2018]. Our algorithm uses stepsize bigger by orders of magnitude. For reader's convenience, we repeat stepsize choices. For AICN stepsize is $\alpha = \frac{-1+\sqrt{1+2G}}{G}$, where $G \stackrel{\text{def}}{=} L_{\text{semi}}\|\nabla f(x_k)\|_{x_k}^*$. For damped Newton methods from Nesterov [2018], $\alpha_1 \stackrel{\text{def}}{=} \frac{1}{1+G_1}, \alpha_2 \stackrel{\text{def}}{=} \frac{1+G_1}{1+G_1+G_1^2}$, where $G_1 \stackrel{\text{def}}{=} L_{\text{sc}}\|\nabla f(x_k)\|_{x_k}^*$.

### A.2    Convergence rate comparison under various assumptions

In this subsection we present Table 2 – comparison of AICN to regularized Newton methods under different smoothness and convexity assumptions.

### A.3    Logistic regression experiments

We solve the following minimization problem:

$$\min_{x \in \mathbb{R}^d} \left\{ f(x) \stackrel{\text{def}}{=} \frac{1}{m} \sum_{i=1}^{m} \log\left(1 - e^{-b_i a_i^\top x}\right) + \frac{\mu}{2}\|x\|_2^2 \right\}.$$

To make problem and data balanced, we normalise each data point and get $\|a_i\|_2 = 1$ for every $i \in [1, \ldots, m]$. Parameters of all methods are fine-tuned to get the fastest convergence. Note, that it is possible that for bigger $L$ method converges faster in practice.

In Figure 4, we consider classification task on dataset *w8a* [Chang and Lin, 2011]. Number of features for every data sample is $d = 300$, $m = 49749$. We take starting point $x_0 \stackrel{\text{def}}{=} 8[1, 1, \ldots, 1]^\top$ and $\mu = 10^{-3}$. Fine-tuned values are $L_{\text{est}} = 0.6$, $L_2 = 0.0001$, $\alpha = 0.5$. We can see that all methods are very close. Damped Newton has rather big step 0.5, so it is fast at the beginning but still struggle at the end because of the fixed step-size.

In Figure 5, we consider binary classification task on dataset *MNIST* [Deng, 2012] (one class contains images with 0, another one — all others). Number of features for every data sample is $d = 28^2 = 784$, $m = 60000$. We take starting point $x_0 \stackrel{\text{def}}{=} 3 \cdot [1, 1, \ldots, 1]^\top$ (such that Newton method started from this point diverges) and $\mu = 10^{-3}$. Fine-tuned values are $L_{\text{est}} = 10$, $L_2 = 0.0003$ for Globally Reg.

Table 2: Convergence guarantees under different versions of convexity and smoothness assumptions. For simplicity, we disregard dependence on bounded level set assumptions. All compared assumptions are considered for $\forall x, h \in \mathbb{R}^d$. We highlight the best know rates in blue.

| Algorithm | Strong convexity constant | Smoothness assumption | Global convergence rate | Local [(1)] convergence exponent | Reference |
|---|---|---|---|---|---|
| Damped Newton B | $0^{(2)}$ | self-concordance (Definition 1) | $\mathcal{O}\left(k^{-\frac{1}{2}}\right)$ | 2 | Nesterov [2018, (5.1.28)] |
| Damped Newton C | $0^{(2)}$ | self-concordance (Definition 1) | ✗ | 2 | Nesterov [2018, (5.2.1)$_C$] |
| Cubic Newton | $\mu$ | Lipschitz-continuous Hessian (4) | $\mathcal{O}\left(k^{-2}\right)$ | 2 | Nesterov and Polyak [2006], Doikov and Nesterov [2022a] |
| Cubic Newton | $\mu$-star-convex | Lipschitz-continuous Hessian (4) | $\mathcal{O}\left(k^{-2}\right)$ | $\frac{3}{2}$ | Nesterov and Polyak [2006] |
| Cubic Newton | non-convex, bounded below | Lipschitz-continuous Hessian (4) | $\mathcal{O}\left(k^{-\frac{2}{3}}\right)$ | ✗ | Nesterov and Polyak [2006] |
| Globally Reg. Newton | $\mu$ | Lipschitz-continuous Hessian (4) | $\mathcal{O}\left(k^{-2}\right)$ | $\frac{3}{2}$ | Mishchenko [4/2021], Doikov and Nesterov [12/2021] |
| Globally Reg. Newton | 0 | Lipschitz-continuous Hessian (4) | $\mathcal{O}\left(k^{-2}\right)$ | ✗ | Mishchenko [4/2021], Doikov and Nesterov [12/2021] |
| **AIC Newton** | $0^{(2)}$ | semi-strong self-concordance (Definition 3) | $\mathcal{O}\left(k^{-2}\right)$ | 2 | Theorems 2, 3 |
| **AIC Newton** | $\mu$ | $f(x+h) - f(x) \leq \langle \nabla f(x), h \rangle + \frac{1}{2}\|h\|_x^2 + \frac{L_{\text{alt}}}{6}\|h\|_x^3$ | $\mathcal{O}\left(k^{-2}\right)$ | 2 | Theorems 4, 3 [(3)] |
| **AIC Newton** | 0 | $f(x+h) - f(x) \leq \langle \nabla f(x), h \rangle + \frac{1}{2}\|h\|_x^2 + \frac{L_{\text{alt}}}{6}\|h\|_x^3$ | $\mathcal{O}\left(k^{-2}\right)$ [(4)] | ✗ | Theorem 4 |

[(1)] For a Lyapunov function $\Phi$ and a constant $c > 0$, we report exponent $\beta > 1$ of $\Phi(x_{k+1}) \leq c\Phi(x_k)^\beta$. Mark ✗ means that such $\beta, c, \Phi$ are not known.

[(2)] Self-concordance implies strong convexity locally.

[(3)] Under strong convexity, we can prove local convergence analogically to Theorem 3.

[(4)] Convergence to a neighborhood of the solution

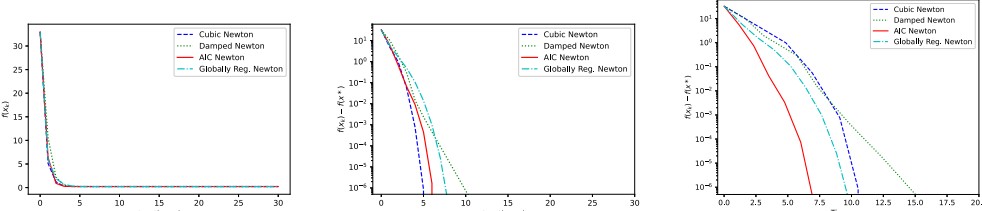

Figure 4: Comparison of regularized Newton methods and Damped Newton method for logistic regression task on *w8a* dataset.

Newton and Cubic Newton, $\alpha = 0.1$. We see that AICN has the same iteration convergence as Cubic Newton but faster by time.

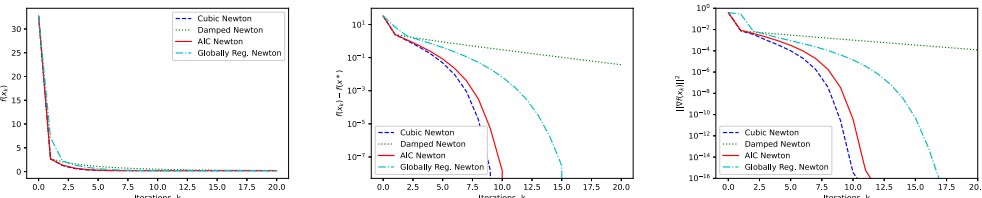

Figure 6: Comparison of regularized Newton methods and Damped Newton method for logistic regression task on *MNIST* dataset (10 models for $i$ vs. other digits problems with argmax aggregation).

In Figure 6, we present the results for multi-class classification problem on dataset *MNIST*. We train 10 different models in parallel, each one for the problem of binary classification distinguishing $i$-th class out of others. Loss on current iteration for the plots is defined as average loss of 10 models.

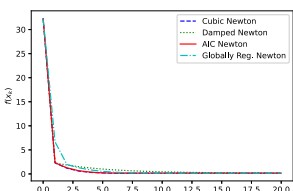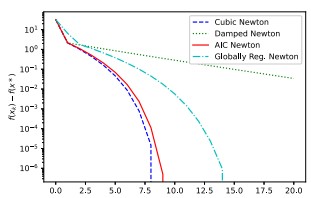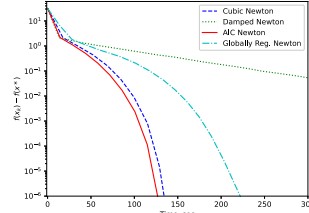

Figure 5: Comparison of regularized Newton methods and Damped Newton method for logistic regression task on *MNIST* dataset (0 vs. all other digits).

Prediction is determined by the maximum "probability" predicted by $i$-th model. The estimates for the parameters of methods are the same as in previous experiment.We see that AICN is the same speed as Cubic Newton and Globally Reg. Newton and much faster than Damped Newton in both value of function and gradient norm.

For normalized problem, we can analytically compute an upper bound for theoretical constant $L_2$

$$\|\nabla^3 f(x)\|_2 \le L_2.$$

One can show that $L_2 = \frac{\sqrt{3}}{18} \simeq 0.1$. In our experiments, we show that Cubic Newton can work with much lower constants: *madelon* - 0.015, *w8a* - 0.00003, *a9a* - 0.000215. It means that theoretical approximation of the constants can be bad and we have to tune them for all methods.

## B    Proofs of Results Appearing in the Paper

In this section, we present proofs of the lemmas and theorems from the main paper's body.

### B.1    Proofs regarding affine invariance (Section 3.1)

*Proof of Lemma 1 (Lemma 5.1.1, Nesterov [2018]).*  **[Newton method is affine-invariant]**

Let $y_k = \mathbf{A}^{-1} x_k$ for some $k \ge 0$ and $\alpha_k$ be affine-invariant. Firstly,

$$y_{k+1} = y_k - \alpha_k \left[\nabla^2 \phi(y_k)\right]^{-1} \nabla \phi(y_k) = y_k - \alpha_k \left[\mathbf{A}^\top \nabla^2 f(\mathbf{A} y_k)\mathbf{A}\right]^{-1} \mathbf{A}^\top \nabla f(\mathbf{A} y_k)$$
$$= \mathbf{A}^{-1} x_k - \alpha_k \mathbf{A}^{-1} \left[\nabla^2 f(x_k)\mathbf{A}\right]^{-1} \nabla f(x_k) = \mathbf{A}^{-1} x_{k+1}.$$

Secondly note that $\|\nabla f(x)\|_x^*$ is affine invariant, as

$$\|\nabla g(y_k)\|_{y_k}^* = \nabla g(y_k)^\top \nabla^2 g(y_k)^{-1} \nabla g(y_k) = \nabla f(x_k)^\top \nabla^2 f(x_k)^{-1} \nabla f(x_k) = \|\nabla f(x_k)\|_{x_k}^*.$$

Consequently, stepsizes $\alpha_k$ from AICN (11) and Nesterov [2018] are all affine-invariant. Hence these damped Newton algorithms are affine-invariant. □

*Proof of Lemma 2.*  **[Upper bound from from semi-strong self-concordance]**

We rewrite function value approximation from the left hand side as

$$f(y) - f(x) - \nabla f(x)[y - x] = \int_0^1 \left(\nabla f(x + \tau(y - x)) - \nabla f(x)\right)[y - x]d\tau$$
$$= \int_0^1 \int_0^\tau \left(\nabla^2 f(x + \lambda(y - x))\right)[y - x]^2 d\lambda d\tau.$$

Taking its norm, we can finish the proof as

$$\left| f(y) - f(x) - \nabla f(x)[y-x] - \frac{1}{2}\nabla^2 f(x)[y-x]^2 \right|$$

$$= \left| \int_0^1 \int_0^\tau \left( \nabla^2 f(x + \lambda(y-x)) - \nabla^2 f(x) \right) [y-x]^2 d\lambda d\tau \right|$$

$$\leq \int_0^1 \int_0^\tau \left| \left( \nabla^2 f(x + \lambda(y-x)) - \nabla^2 f(x) \right) [y-x]^2 \right| d\lambda d\tau$$

$$\overset{(14)}{\leq} \int_0^1 \int_0^\tau L_{\text{semi}} \lambda \|y-x\|_x^3 d\lambda d\tau = \frac{L_{\text{semi}}}{6} \|y-x\|_x^3.$$

$\square$

*Proof of Theorem [1].* [**Minimizer of the model ([18]) has form of damped Newton method ([11])**]

Proof is straightforward. To show that AICN model update minimizes $S_{f,L_{\text{est}}}(x)$, we compute the gradient of $S_{f,L_{\text{est}}}(x)$ at next iterate of AICN. Showing that it is 0 concludes $x_{k+1} = S_{f,L_{\text{est}}}(x_k)$.

For simplicity, denote $h \overset{\text{def}}{=} y - x$. We can simplify the implicit update step $S_{f,L_{\text{est}}}(x)$

$$S_{f,L_{\text{est}}}(x) = \underset{y \in \mathbb{E}}{\operatorname{argmin}} \left\{ f(x) + \langle \nabla f(x), y-x \rangle + \frac{1}{2} \langle \nabla^2 f(x)(y-x), y-x \rangle + \frac{L_{\text{est}}}{6} \|y-x\|_x^3 \right\} \tag{29}$$

$$= x + \underset{h \in \mathbb{E}}{\operatorname{argmin}} \left\{ \langle \nabla f(x), h \rangle + \frac{1}{2} \|h\|_x^2 + \frac{L_{\text{est}}}{6} \|h\|_x^3 \right\}. \tag{30}$$

Taking gradient of the subproblem with respect to $h$,

$$\nabla_h \left( \langle \nabla f(x), h \rangle + \frac{1}{2} \|h\|_x^2 + \frac{L_{\text{est}}}{6} \|h\|_x^3 \right) = \nabla f(x) + \nabla^2 f(x) h + \frac{L_{\text{est}}}{2} \nabla^2 f(x) h \|h\|_x. \tag{31}$$

and setting $h$ according to AICN, $h = -\alpha \nabla^2 f(x)^{-1} \nabla f(x)$, leads to

$$\nabla f(x) - \alpha \nabla f(x) - \frac{L_{\text{est}}}{2} \alpha^2 \nabla f(x) \left\| \nabla^2 f(x)^{-1} \nabla f(x) \right\|_x = -\nabla f(x) \left( -1 + \alpha + \frac{L_{\text{est}}}{2} \alpha^2 \|\nabla f(x)\|_x^* \right). \tag{32}$$

Finally, AICN stepsize $\alpha$ ([11]) is chosen as a root of quadratic function

$$\frac{L_{\text{est}}}{2} \|\nabla f(x)\|_x^* \alpha^2 + \alpha - 1 = 0, \tag{33}$$

hence the gradient of the ([17]) at next iterate of AICN is 0. This concludes the proof. $\square$

## B.2 Proof of global convergence (Section [4.1])

*Proof of Lemma [3].* [**One step decrease globally under semi-strong self-concordance**]

This claim follows directly from Lemma [2].

$$f(S_{f,L_{\text{est}}}(x)) \overset{(16)}{\leq} f(x) + \langle \nabla f(x), S_{f,L_{\text{est}}}(x) - x \rangle + \frac{1}{2} \langle \nabla^2 f(x)(S_{f,L_{\text{est}}}(x) - x), S_{f,L_{\text{est}}}(x) - x \rangle$$

$$+ \frac{L_{\text{est}}}{6} \|S_{f,L_{\text{est}}}(x) - x\|_x^3$$

$$\overset{(17)}{=} \min_{y \in \mathbb{E}} \left\{ f(x) + \langle \nabla f(x), y-x \rangle + \frac{1}{2} \langle \nabla^2 f(x)(y-x), y-x \rangle + \frac{L_{\text{est}}}{6} \|y-x\|_x^3 \right\}$$

$$\overset{(15)}{\leq} \min_{y \in \mathbb{E}} \left\{ f(y) + \frac{L_{\text{est}}}{3} \|y-x\|_x^3 \right\}.$$

$\square$

*Proof of Theorem* 2. **[Global convergence under semi-strong self-concordance]**

We start by taking Lemma 3 for any $t \geq 0$, we obtain

$$
\begin{aligned}
f(x_{t+1}) &\overset{(22)}{\leq} \min_{y \in \mathbb{E}} \left\{ f(y) + \frac{L_{\text{est}}}{3} \|y - x_t\|_{x_t}^3 \right\} \\
&\overset{(23)}{\leq} \min_{\eta_t \in [0,1]} \left\{ f(x_t + \eta_t(x_* - x_t)) + \frac{L_{\text{est}}}{3} \eta_t^3 D^3 \right\} \\
&\leq \min_{\eta_t \in [0,1]} \left\{ (1 - \eta_t) f(x_t) + \eta_t f(x_*) + \frac{L_{\text{est}}}{3} \eta_t^3 D^3 \right\},
\end{aligned}
$$

where for the second line we take $y = x_t + \eta_t(x_* - x_t)$ and use convexity of $f$ for the third line. Therefore, subtracting $f(x_*)$ from both sides, we obtain, for any $\eta_t \in [0, 1]$

$$
f(x_{t+1}) - f(x_*) \leq (1 - \eta_t)(f(x_t) - f(x_*)) + \frac{L_{\text{est}}}{3} \eta_t^3 D^3 \tag{34}
$$

Let us define the sequence $\{A_t\}_{t \geq 0}$ as follows:

$$
A_t \overset{\text{def}}{=} \begin{cases} 1, & t = 0 \\ \displaystyle\prod_{i=1}^{t}(1 - \eta_i), & t \geq 1. \end{cases} \tag{35}
$$

Then $A_t = (1 - \eta_t)A_{t-1}$. Also, we define $\eta_0 \overset{\text{def}}{=} 1$. Dividing both sides of (34) by $A_t$, we get

$$
\begin{aligned}
\frac{1}{A_t}(f(x_{t+1}) - f(x_*)) &\leq \frac{1}{A_t}(1 - \eta_t)(f(x_t) - f(x_*)) + \frac{\eta_t^3}{A_t} \frac{L_{\text{est}} D^3}{3} \\
&= \frac{1}{A_{t-1}}(f(x_t) - f(x_*)) + \frac{\eta_t^3}{A_t} \frac{L_{\text{est}} D^3}{3}. \tag{36}
\end{aligned}
$$

Summing both sides of inequality (36) for $t = 0, \ldots, k$, we obtain

$$
\begin{aligned}
\frac{1}{A_k}(f(x_{k+1}) - f(x_*)) &\leq \frac{(1 - \eta_0)}{A_0}(f(x_0) - f(x_*)) + \frac{L_{\text{est}} D^3}{3} \sum_{t=0}^{k} \frac{\eta_t^3}{A_t} \\
&\overset{\eta_0 = 1}{=} \frac{L_{\text{est}} D^3}{3} \sum_{t=0}^{k} \frac{\eta_t^3}{A_t}.
\end{aligned}
$$

As a result,

$$
f(x_{k+1}) - f(x_*) \leq \frac{L_{\text{est}} D^3}{3} \sum_{t=0}^{k} \frac{A_k \eta_t^3}{A_t} p \tag{37}
$$

To finish the proof, we need to choose $\eta_t$ so that $\sum_{t=0}^{k} \frac{A_k \eta_t^3}{A_t} = O(k^{-2})$. This holds for[11]

$$
\eta_t \overset{\text{def}}{=} \frac{3}{t+3}, \quad t \geq 0, \tag{38}
$$

as stated in the next lemma.

**Lemma 6** (Properties of $\eta_t$ and $A_t$, [from (2.23), Ghadimi et al. [2017]). ] *For choice $\eta_t$ as (38) and $A_t$ as (35) we have*

$$
A_t = \frac{6}{(t+1)(t+2)(t+3)}, \tag{39}
$$

$$
\sum_{t=1}^{k} \frac{\eta_t^3}{A_t} = \sum_{t=1}^{k} \frac{9(t+1)(t+2)}{2(t+3)^2} \leq \frac{9k}{2}. \tag{40}
$$

---

[11]Note that formula of $\eta_0$ coincides with its definition above.

Plugging Lemma 6 inequalities to (37) concludes the proof of the Theorem 2,

$$f(x_{k+1}) - f(x_*) \leq \frac{6}{(k+1)(k+2)(k+3)} \frac{L_{\text{est}}D^3}{3} \frac{9k}{2} \leq \frac{9L_{\text{est}}D^3}{k^2} \leq \frac{9L_{\text{est}}R^3}{k^2}.$$

$\square$

For readers convenience, we include proof of Lemma 6 in Appendix B.4.

### B.3 Proofs of local convergence (Section 4.2)

*Proof of Lemma 4.* **[Strong convexity / Bound on inverse Hessian norm change]**

Claim follows from Theorem 5.1.7 of Nesterov [2018], which states that for $L_{\text{sc}}$-self-concordant function, hence also for $L_{\text{semi}}$-semi-strongly self-concordant function $f$ and $x_k, x_{k+1}$ such that $\frac{L_{\text{semi}}}{2}\|x_{k+1} - x_k\|_{x_k} < 1$ holds

$$\left(1 - \frac{L_{\text{semi}}}{2}\|x_{k+1} - x_k\|_{x_k}\right)^2 \nabla^2 f(x_{k+1}) \preceq \nabla^2 f(x_k) \preceq \left(1 - \frac{L_{\text{semi}}}{2}\|x_{k+1} - x_k\|_{x_k}\right)^{-2} \nabla^2 f(x_{k+1}).$$

Let $c$ be some constant, $0 < c < 1$. Then for updates of AICN in the neighborhood

$$\left\{x_k : c \geq \frac{L_{\text{est}}}{2}\alpha_k\|\nabla f(x_k)\|_{x_k}^*\right\}$$

holds

$$\nabla^2 f(x_{k+1})^{-1} \preceq (1-c)^{-2} \nabla^2 f(x_k)^{-1}. \tag{41}$$

$\square$

In order to prove Lemma 5, we first use semi-strong self-concordance to prove a key inequality – a version of Hessian smoothness, bounding gradient approximation by difference of points.

**Lemma 7.** *For semi-strongly self-concordant function $f$ holds*

$$\left\|\nabla f(y) - \nabla f(x) - \nabla^2 f(x)[y - x]\right\|_x^* \leq \frac{L_{semi}}{2}\|y - x\|_x^2. \tag{42}$$

*Proof of Lemma 7.* **[Local smoothness assumption follows from semi-strong self-concordance]**

We rewrite gradient approximation on the left hand side as

$$\nabla f(y) - \nabla f(x) - \nabla^2 f(x)[y - x] = \int_0^1 \left(\nabla^2 f(x + \tau(y - x)) - \nabla^2 f(x)\right)[y - x]d\tau.$$

Now, we can bound it in dual norm as

$$\left\|\nabla f(y) - \nabla f(x) - \nabla^2 f(x)[y - x]\right\|_x^* = \left\|\int_0^1 \left(\nabla^2 f(x + \tau(y - x)) - \nabla^2 f(x)\right)[y - x]d\tau\right\|_x^*$$

$$\leq \int_0^1 \left\|\left(\nabla^2 f(x + \tau(y - x)) - \nabla^2 f(x)\right)[y - x]\right\|_x^* d\tau$$

$$\leq \int_0^1 \left\|\nabla^2 f(x + \tau(y - x)) - \nabla^2 f(x)\right\|_{op}\|y - x\|_x d\tau$$

$$\overset{(14)}{\leq} \int_0^1 L_{\text{semi}}\tau\|y - x\|_x^2 d\tau = \frac{L_{\text{semi}}}{2}\|y - x\|_x^2,$$

where in second inequality we used property of operator norm (10). $\square$

Finally, we are ready to prove one step decrease and the convergence theorem.

*Proof of Lemma 5.* **[One step decrease locally under semi-strong self-concordance]**

We bound norm of $\nabla f(x_{k+1})$ using basic norm manipulation and triangle inequality as

$$\|\nabla f(x_{k+1})\|_{x_k}^* \overset{(11)}{=} \left\|\nabla f(x_{k+1}) - \nabla^2 f(x_k)(x_{k+1} - x_k) - \alpha_k \nabla f(x_k)\right\|_{x_k}^*$$

$$= \left\|\nabla f(x_{k+1}) - \nabla f(x_k) - \nabla^2 f(x_k)(x_{k+1} - x_k) + (1 - \alpha_k)\nabla f(x_k)\right\|_{x_k}^*$$

$$\leq \left\|\nabla f(x_{k+1}) - \nabla f(x_k) - \nabla^2 f(x_k)(x_{k+1} - x_k)\right\|_{x_k}^* + (1 - \alpha_k)\|\nabla f(x_k)\|_{x_k}^*$$

Using Lemma 7, we can continue

$$\|\nabla f(x_{k+1})\|_{x_k}^* \leq \left\|\nabla f(x_{k+1}) - \nabla f(x_k) - \nabla^2 f(x_k)(x_{k+1} - x_k)\right\|_{x_k}^* + (1 - \alpha_k)\|\nabla f(x_k)\|_{x_k}^*$$

$$\overset{(42)}{\leq} \frac{L_{\text{semi}}}{2}\|x_{k+1} - x_k\|_{x_k}^2 + (1 - \alpha_k)\|\nabla f(x_k)\|_{x_k}^*$$

$$\overset{(11)}{\leq} \frac{L_{\text{semi}}\alpha_k^2}{2}\|\nabla f(x_k)\|_{x_k}^{*2} + (1 - \alpha_k)\|\nabla f(x_k)\|_{x_k}^*$$

$$\leq \frac{L_{\text{est}}\alpha_k^2}{2}\|\nabla f(x_k)\|_{x_k}^{*2} + (1 - \alpha_k)\|\nabla f(x_k)\|_{x_k}^*$$

$$= \left(\frac{L_{\text{est}}\alpha_k^2}{2}\|\nabla f(x_k)\|_{x_k}^* - \alpha_k + 1\right)\|\nabla f(x_k)\|_{x_k}^*$$

$$\overset{(33)}{=} L_{\text{est}}\alpha_k^2\|\nabla f(x_k)\|_{x_k}^{*2}$$

We use Lemma 4 to shift matrix norms.

$$\|\nabla f(x_{k+1})\|_{x_{k+1}}^* \overset{(41)}{\leq} \frac{1}{1-c}\|\nabla f(x_{k+1})\|_{x_k}^*$$

$$\overset{(26)}{\leq} \frac{L_{\text{est}}\alpha_k^2}{1-c}\|\nabla f(x_k)\|_{x_k}^{*2} \qquad (43)$$

$$< \frac{L_{\text{est}}\alpha_k}{1-c}\|\nabla f(x_k)\|_{x_k}^{*2}.$$

We obtain neighborhood of decrease by solving $\frac{L_{\text{est}}\alpha_k}{1-c}\|\nabla f(x_k)\|_{x_k}^* \leq 1$ in $\|\nabla f(x_k)\|_{x_k}^*$.

$\square$

*Proof of Theorem 3.* **[Local convergence under semi-strong self-concordance]**

Let $c = \frac{1}{3}$, then for $\|\nabla f(x_0)\|_{x_0}^* < \frac{8}{9L_{\text{est}}}$, we have $\frac{L_{\text{est}}\alpha_0}{1-c}\|\nabla f(x_0)\|_{x_0}^* \leq 1$ and $c \geq \frac{L_{\text{est}}}{2}\alpha_0\|\nabla f(x_0)\|_{x_0}^*$. Then from Lemma 5 we have guaranteed the decrease of gradients $\|g_{k+1}\|_{x_{k+1}}^* \leq \|g_k\|_{x_k}^* < \frac{8}{9L_{\text{est}}}$. We finish proof by chaining (43) and simplifying with $\alpha_i \leq 1$.

$$\|\nabla f(x_k)\|_{x_k}^* \leq \left(\tfrac{3}{2}L_{\text{est}}\right)^k \left(\prod_{i=0}^{k}\alpha_i^2\right)\left(\|\nabla f(x_0)\|_{x_0}^*\right)^{2^k}. \qquad (44)$$

$\square$

## B.4 Technical lemmas

**Lemma 8** (Arithmetic mean – Geometric mean inequality)**.** *For $c \geq 0$ we have*

$$1 + c = \frac{1 + (1 + 2c)}{2} \overset{AG}{\geq} \sqrt{1 + 2c}. \qquad (45)$$

**Lemma 9** (Jensen for square root)**.** *Function $f(x) = \sqrt{x}$ is concave, hence for $c \geq 0$ we have*

$$\frac{1}{\sqrt{2}}(\sqrt{c} + 1) \leq \sqrt{c+1} \qquad \leq \sqrt{c} + 1. \qquad (46)$$

**Lemma 6** [(2.23) from [Ghadimi et al., 2017]] For

$$\eta_t \stackrel{\text{def}}{=} \frac{3}{t+3}, \quad t \geq 0, \qquad \text{and} \qquad A_t \stackrel{\text{def}}{=} \begin{cases} 1, & t = 0 \\ \prod_{i=1}^{t}(1-\eta_i), & t \geq 1 \end{cases}$$

we have

$$A_t = \frac{6}{(t+1)(t+2)(t+3)} \qquad \text{and} \qquad \sum_{t=1}^{k} \frac{\eta_k^3}{A_t} = \sum_{t=1}^{k} \frac{9(t+1)(t+2)}{2(t+3)^2} \leq \frac{3k}{2}. \tag{47}$$

*Proof of Lemma 6.* We have

$$A_k = \prod_{t=1}^{k}(1-\eta_t) = \prod_{t=1}^{k} \frac{t}{t+3} = \frac{k!\,3!}{(k+3)!} = 3! \prod_{j=1}^{3} \frac{1}{k+j}, \tag{48}$$

which gives,

$$\sum_{t=0}^{k} \frac{A_k \eta_t^3}{A_t} = \sum_{t=0}^{T} \frac{3^3}{(t+3)^3} \prod_{j=1}^{3} \frac{t+j}{k+j} = 3^3 \prod_{j=1}^{3} \frac{1}{k+j} \sum_{t=0}^{k} \frac{\prod_{j=1}^{3}(t+j)}{(t+3)^3}. \tag{49}$$

The sum is non-decreasing. Indeed, we have

$$1 \leq 1 + \frac{1}{t+3} \leq 1 + \frac{1}{t+j}, \quad \forall j \in \{1,2,3\},$$

and, hence, for all $j \in \{1,2,3\}$,

$$\left(1 + \tfrac{1}{t+3}\right)^3 \leq \prod_{j=1}^{3}\left(1 + \frac{1}{t+j}\right)$$

$$\Leftrightarrow \quad \left(\tfrac{t+4}{t+3}\right)^3 \leq \prod_{j=1}^{3} \frac{t+j+1}{t+j}$$

$$\Leftrightarrow \quad \frac{\prod_{j=1}^{3}(t+j)}{(t+3)^3} \leq \frac{\prod_{j=1}^{3}(t+1+j)}{(t+1+3)^3}.$$

Thus, we have shown that the summands in the RHS of (49) are growing, whence we get the next upper bound for the sum

$$\begin{aligned} \sum_{t=0}^{k} \frac{A_k \eta_t^3}{A_t} &= 3^3 \prod_{j=1}^{3} \frac{1}{k+j} \sum_{t=0}^{k} \frac{\prod_{j=1}^{3}(t+j)}{(t+3)^3} \\ &\leq 3^3 \prod_{j=1}^{3} \frac{1}{k+j} \cdot (k+1) \cdot \frac{\prod_{j=1}^{3}(k+j)}{(k+3)^3} \leq \frac{(k+1)3^3}{(k+3)^3} \leq O\left(\frac{1}{k^2}\right). \end{aligned} \tag{50}$$

$\square$

## C   Global Convergence with weaker assumptions on Self-Concordance

We can prove global convergence to a neighborhood of the solution without using self-concordance directly, just by utilizing the following assumptions:

**Assumption 2** (Convexity). *For function $f$ and any $x, h \in \mathbb{E}$ holds*

$$f(x+h) \geq f(x) + \langle \nabla f(x), h \rangle \tag{51}$$

**Assumption 3** (Hessian smoothness, in Hessian norms). *Objective function $f$ satisfy*

$$f(x+h) - f(x) \leq \langle \nabla f(x), h \rangle + \tfrac{1}{2}\|h\|_x^2 + \tfrac{L_{alt}}{6}\|h\|_x^3, \qquad \forall x, h \in \mathbb{E}. \tag{52}$$

**Lemma 10** (One step decrease globally)**.** *Let Assumption 3 hold and let $L_{est} \geq L_{alt}$. Iterates of* AICN *eq. (11) yield function value decrease,*

$$
f(x_{k+1}) - f(x_k) \leq \begin{cases} -\frac{1}{2\sqrt{L_{est}}}\|\nabla f(x_k)\|_{x_k}^{*\frac{3}{2}} & \text{if } \|\nabla f(x_k)\|_{x_k}^{*} \geq \frac{4}{L_{est}} \\ -\frac{1}{4}\|\nabla f(x_k)\|_{x_k}^{*2} & \text{if } \|\nabla f(x_k)\|_{x_k}^{*} \leq \frac{4}{L_{est}} \\ -\frac{\sqrt{c_1}}{2\sqrt{L_{est}}}\|\nabla f(x_k)\|_{x_k}^{*\frac{3}{2}} & \text{if } \|\nabla f(x_k)\|_{x_k}^{*} \geq \frac{4c_1}{L_{est}} \text{ and } 0 < c_1 \leq 1 \end{cases} . \tag{53}
$$

Decrease of Lemma 10 is tight up to a constant factor. As far as $\|\nabla f(x_k)\|_{x_k}^{*} \leq \frac{4c_1}{L_{est}}$, we have functional value decrease as the first line of (53), which leads to $\mathcal{O}\left(k^{-2}\right)$ convergence rate.

We can obtain fast convergence to only neighborhood of solution, because close to the solution, gradient norm is sufficiently small $\|\nabla f(x_k)\|_{x_k}^{*} \leq \frac{4c_1}{L_{est}}$ and we get functional value decrease from second line of (53). However, this convergence is slower then $\mathcal{O}\left(k^{-2}\right)$ for $\|\nabla f(x_k)\|_{x_k}^{*} \approx 0$ and it is insufficient for $\mathcal{O}(k^{-2})$ rate.

Note that third line generalizes first line; we use it to remove a constant factor gap from the neighborhood of fast local convergence.

**Theorem 4** (Global convergence)**.** *Let Assumptions 2, 3, 1 hold, and constants $c_1, L_{est}$ satisfy $0 < c_1 \leq 1, L_{est} \geq L_{alt}$. For $k$ until $\|\nabla f(x_k)\|_{x_k}^{*} \geq \frac{4c_1}{L_{est}}$,* AICN *has global quadratic decrease,*
$$
f(x_k) - f^* \leq \mathcal{O}\left(\frac{L_{est} D^3}{k^2}\right).
$$

Note that the global quadratic decrease of AICN is only to a neighborhood of the solution. However, once AICN gets to this neighborhood, it converges using (faster) local convergence rate (Theorem 3).

**Proofs of global convergence without self-concordance**

Throughout the rest of proofs, we simplify expressions by denoting terms

$$
g_k \overset{\text{def}}{=} \nabla f(x_k) \qquad \text{and} \qquad h_k \overset{\text{def}}{=} x_{k+1} - x_k , \tag{54}
$$

for which holds

$$
h_k = -\alpha_k \nabla^2 f(x_k)^{-1} g_k, \quad g_k = -\frac{1}{\alpha_k}\nabla^2 f(x_k) h_k \quad \text{and} \quad \|h_k\|_{x_k} = \sqrt{\alpha}\|g_k\|_{x_k}^{*} m \tag{55}
$$

and also $G_k \overset{\text{def}}{=} L_{est}\|g_k\|_{x_k}^{*}.$

*Proof of Lemma 10.* We can use Assumption 3 to obtain

$$
\begin{aligned}
f(x_{k+1}) - f(x_k) &= f(x_k + h_k) - f(x_k) \\
&\overset{(52)}{\leq} \langle \nabla f(x_k), h_k \rangle + \frac{1}{2}\|h_k\|_{x_k}^2 + \frac{L_{est}}{6}\|h_k\|_{x_k}^3 \\
&\overset{(55)}{=} -\alpha_k\|g_k\|_{x_k}^{*2} + \frac{1}{2}\alpha_k^2\|g_k\|_{x_k}^{*2} + \frac{L_{est}}{6}\alpha_k^3\|g_k\|_{x_k}^{*3} \\
&= -\alpha_k\|g_k\|_{x_k}^{*2}\left(1 - \frac{1}{2}\alpha_k - \frac{L_{est}}{6}\alpha_k^2\|g_k\|_{x_k}^{*}\right).
\end{aligned} \tag{56} \tag{57}
$$

We can simplify bracket in eq. (57) as

$$
\begin{aligned}
1 - \frac{1}{2}\alpha_k - \frac{L_{est}}{6}\alpha_k^2\|g_k\|_{x_k}^{*} &= 1 - \frac{1}{2}\frac{\sqrt{1+2G_k}-1}{G_k} - \frac{G_k}{6}\left(\frac{\sqrt{1+2G_k}-1}{G_k}\right)^2 \\
&= \frac{4G_k + 1 - \sqrt{1+2G_k}}{6G_k} \overset{(45)}{\geq} \frac{1}{2}.
\end{aligned}
$$

Also, we can simplify the other term in eq. (57) as

$$\alpha_k \|g_k\|_{x_k}^{*2} = \frac{\|g_k\|_{x_k}^*}{L_{\text{est}}} \left( \sqrt{1 + 2G_k} - 1 \right) \frac{\sqrt{1 + 2G_k} + 1}{\sqrt{1 + 2G_k} + 1} = \frac{2\|g_k\|_{x_k}^{*2}}{\sqrt{1 + 2G_k} + 1}$$

$$\overset{(46)}{\geq} \frac{2\|g_k\|_{x_k}^{*2}}{\sqrt{G_k} + 1 + \frac{1}{\sqrt{2}}} \geq \frac{2\|g_k\|_{x_k}^{*2}}{\sqrt{G_k} + 2} \geq \frac{\|g_k\|_{x_k}^{*2}}{\max\left(\sqrt{G_k}, 2\right)},$$

and plug these two result into eq. (57) to obtain first two lines of (53). Third line can be obtained from the first line of (53). For $c_1$ so that $0 < c_1 \leq 1$ and $x_k$ satisfying $\frac{4c_1}{L_{\text{est}}} \leq \|\nabla f(x_k)\|_{x_k}^* < \frac{4}{L_{\text{est}}}$ holds

$$f(x_{k+1}) - f(x_k) \leq -\frac{1}{4}\|\nabla f(x_k)\|_{x_k}^{*2} \leq -\frac{\sqrt{c_1}}{2\sqrt{L_{\text{est}}}}\|\nabla f(x_k)\|_{x_k}^{*\frac{3}{2}}.$$

$\square$

*Proof of Theorem 4.* As a consequence of convexity (Assumption 2) and bounded level sets (Assumption 1), we have

$$f(x_k) - f^* \leq \langle g_k, x_k - x_* \rangle = \left\langle \nabla^2 f(x_k)^{-1/2} g_k, \nabla^2 f(x_k)^{1/2}(x_k - x_*) \right\rangle \leq \|g_k\|_{x_k}^* \|x_k - x_*\|_{x_k}$$

$$\leq D\|g_k\|_{x_k}^*. \tag{58}$$

Plugging it into eq. (53) (which holds under Assumption 3) and noting that it yields

$$f(x_{k+1}) - f(x_k) \leq -\frac{\sqrt{c_1}}{2\sqrt{L_{\text{est}}}D^{3/2}} \left( f(x_k) - f^* \right)^{3/2}. \tag{59}$$

Denote $\tau \overset{\text{def}}{=} \frac{\sqrt{c_1}}{2\sqrt{L_{\text{est}}}D^{3/2}}$ and $\beta_k \overset{\text{def}}{=} \tau^2(f(x_k) - f^*) \geq 0$. Terms $\beta_k$ satisfy recurrence

$$\beta_{k+1} = \tau^2(f(x_{k+1}) - f^*) \overset{(53)}{\leq} \tau^2(f(x_k) - f^*) - \tau^3(f(x_k) - f^*)^{3/2} = \beta_k - \beta_k^{3/2}.$$

Because $\beta_{k+1} \geq 0$, we have that $\beta_k \leq 1$.

Proposition 1 of Mishchenko [4/2021] shows that the sequence $\{\beta_k\}_{k=0}^\infty, 0 \leq \beta_k \leq 1$ decreases as $\mathcal{O}\left(\frac{1}{k^2}\right)$. Let $c_2$ be a constant satisfying $\beta_k \leq \frac{c_2}{k^2}$ for all $k$ ($c_2 \approx 3$ seems to be sufficient). Finally, fol $k \geq \frac{\sqrt{c_2}}{\tau\sqrt{\varepsilon}} = 2\sqrt{\frac{c_2 L_{\text{est}}D^3}{c_1\varepsilon}} = \mathcal{O}\left(\sqrt{\frac{L_{\text{est}}D^3}{\varepsilon}}\right)$ we have

$$f(x_k) - f^* = \frac{\beta_k}{\tau^2} \leq \frac{c_2}{c_1 k^2 \tau^2} \leq \varepsilon.$$

$\square$