# OpenReview forum: "A Damped Newton Method Achieves Global $\mathcal O \left(\frac{1}{k^2}\right)$  and Local Quadratic  Convergence Rate"
_NeurIPS.cc/2022/Conference — NeurIPS 2022 Accept_

### Official Review · Reviewer_dYEq · 2022-06-23

**Rating:** 8
**Confidence:** 3
**Soundness:** 4 excellent
**Presentation:** 4 excellent
**Contribution:** 4 excellent

**Summary:**

The paper provide a Newton Method with simple step size and provide an affine-invariance analysis of it with local and global convergence rates.

**Questions:**

None.

**Limitations:**

None.

**Strengths And Weaknesses:**

Pros:
- The paper is very well-written and easy to follow.
- The contributions are elegant and significant.
- There are compelling numerical experiment.

CCL: this is the best paper in my batch of review.

---

> ### Author Response · Authors · 2022-08-02
> **Response to review**
>
> We would like to thank the reviewer for the very positive review. We are happy to hear you like our paper.

---

### Official Review · Reviewer_42nd · 2022-06-27

**Rating:** 6
**Confidence:** 4
**Soundness:** 3 good
**Presentation:** 3 good
**Contribution:** 3 good

**Summary:**

It is well known that Newton's method can fail to converge even for smooth, convex functions. This paper proposes a damped Newton's method with a stepsize schedule such that the method achieves $\mathcal{O}(1/k^2)$ global rate, which matches the convergence rate of Nesterov's cubic regularization method. The main idea is to minimize a cubic upper bound similar to the one used in cubic regularization, but replace the cubic term with a local Hessian norm.
Compared to previous methods, the proposed algorithm has an explicit formula for the step-size and enjoys fast local and global convergence.

**Questions:**

- As the authors claim,  one advantage of AIC Newton is that it is affine-invariant. This is indeed a natural consequence of replacing the $\ell_2$ norm by the local Hessian norm in the cubic term. But i'm not sure what is the real advantage here in terms of practical performance? What is the actual benefit of the method satisfying self-concordance?
- In the numerical experiments, the authors say that the parameters $L_f, L_2$ and $\alpha$ are all fine-tuned. How is this done exactly? In general, how would I pick $L_f$ for AIC Newton?
- In terms of per-iteration cost, does AIC Newton offer any advantage compared to cubic Newton of Nesterov? There has been many recent work that reduce the cost of the cubic subproblem [1] [2]. In light of these, how does of real running time of AIC newton compare with cubic Newton?


[1] Carmon, Y., & Duchi, J. (2019). Gradient descent finds the cubic-regularized nonconvex Newton step. SIAM Journal on Optimization, 29(3), 2146-2178.

[2] Cartis, C., Gould, N. I., & Toint, P. L. (2011). Adaptive cubic regularisation methods for unconstrained optimization. Part I: motivation, convergence and numerical results. Mathematical Programming, 127(2), 245-295.

**Limitations:**

Yes.

**Strengths And Weaknesses:**

This article is well-written and offers an overview of potential problems for Newton's method and how these problems were previously addressed. The authors motivate their algorithm, which they call AIC newton, quite naturally and the idea behind the proofs are also nicely communicated. In the end, under certain assumptions, the proposed algorithm is easy to implement and has strong local and global guarantees. However, there are a few issues that prevent me from giving this article a higher score.

 1. The assumptions under which convergence is proved are quite strong. In particular, Theorem 2, which guarantees global convergence, requires that the function is weakly-strongly self-conconrdant and also *convex*. Compare this with cubic newton of Nesterov, which guarantees convergence towards a second-order stationary point for *non-convex* functions at the same rate, only under the assumption of gradient and Hessian lipschitz. The authors does indeed consider a relaxation of the self-concordance assumption, but in the case the convergence rate the prove is slower, and does not match best known results.

2. The author states that one key advantage of the proposed algorithm is that it has an explicit stepsize and does not require line search, like previous methods. It seems to me that this claim is inaccurate, since computing the step-size requires knowledge of the parameter $L\_f$, which is a constant related to self-concordance. In cubic regularization, for instance, if the Hessian lipschitz constant is also known, then the line search can also be avoided. Therefore, in this regard the proposed method again offers no advantage over cubic newton.

3. In the numerical experiments, it seems to me that cubic Newton often outperforms the AIC newton, unlike what the authors claim in the intro. However the difference is usually minor and i do not consider this an huge issue.

In the end, I do think the global convergence of Newton's method is important and interesting for the NeurIPS community. The proposed method, AIC newton can potentially be a useful algorithm, but my main issue is that it has restrictive assumptions and no clear advantage over cubic newton.

---

> ### Author Response · Authors · 2022-08-02
> **Response to review (part 1)**
>
> >**Weakness 1:** *The assumptions under which convergence is proved are quite strong. In particular, Theorem 2, which guarantees global convergence, requires that the function is weakly-strongly self-concordant and also convex. Compare this with cubic newton of Nesterov, which guarantees convergence towards a second-order stationary point for non-convex functions at the same rate, only under the assumption of gradient and Hessian lipschitz. The authors does indeed consider a relaxation of the self-concordance assumption, but in the case the convergence rate the prove is slower, and does not match best known results.*
>
> **Response:**
> We can elaborate on the comparison to Cubic Newton under different assumptions:
> - For convex functions, both AICN and Cubic Newton converge at rate $O(k^{-2})$.
> - For general non-convex functions, the convergence rate of Cubic Newton to the local minimum is $O(k^{-2/3})$ (see Theorem 1 in [1], this matches the lower bound from [2]).
> Convergence of AICN on the general non-convex problems wasn’t investigated - it was out of the scope of our project. It is a direction for future research.
> - In the original paper [1], the authors showed that Cubic Newton converges with rate $O(k^{-2})$ for some specific non-convex problems only (e.g, star-convex functions).
>
> Analyzing AICN under those relaxed assumptions was out of the scope of the paper; it is one of the future directions of research.
>
> ---
> >**Weakness 2:** *The author states that one key advantage of the proposed algorithm is that it has an explicit stepsize and does not require line search, like previous methods. It seems to me that this claim is inaccurate, since computing the step-size requires knowledge of the parameter Lf, which is a constant related to self-concordance. In cubic regularization, for instance, if the Hessian lipschitz constant is also known, then the line search can also be avoided. Therefore, in this regard the proposed method again offers no advantage over cubic newton.*
>
> **Response:** We believe this to be a misunderstanding, we would like to clarify it. There are **two** possible line-searches that can be considered while running Cubic Newton: **(A)** line-search to choose a stepsize / estimate the smoothness constant $L_2$, **(B)** line-search to solve the subproblem in iteration $k$ (to obtain $x^{k+1}$ from $x^k$).
> We would like to clarify that throughout our whole paper we are comparing non-adaptive methods (AICN method to non-adaptive Cubical Newton), hence we don’t talk about (A). When we claim we can avoid line-search, we mean we can avoid line-search (B).
>
> To elaborate on part (A), we assume that we know $L_f$ (or its approximation) for AICN. Authors of Cubic Newton assume knowledge of $L_2$ (or its approximation). Those assumptions are of similar strength. In practice, both of these parameters can be treated as hyperparameters.
> To elaborate on (B), Cubic Newton with known $L_2$ needs **additional** line-search to solve a certain key subproblem in each iteration. See Section 5.1 in the original paper [1], or Subsection 4.1.4.1 in [3].
>
> ---
>
> >**Weakness 3:** *In the numerical experiments, it seems to me that cubic Newton often outperforms the AIC newton, unlike what the authors claim in the intro. However the difference is usually minor and i do not consider this an huge issue.*
>
> **Response:** Yes, AICN and Cubic Newton are very close in the final version of experiments. Sometimes AIC is faster (Figure 1, 4), sometimes Cubic Newton ios faster (Figure 2,3). Please note that we report iteration vs loss. These plots are disadvantageous for AICN as they disregard the cheaper iteration complexity of AICN $O(d^3)$ compared to CN $O(d^3 \log \varepsilon^{-1})$. This comes from the fact that Cubic Newton has a subproblem that requires line-search and inverting the Hessian matrix $O(\log \varepsilon^{-1})$ times in each iteration. For more details, you can check Subsection 4.1.4 of [3] or the original paper about Cubic Newton [1].
>
> We chose to use this disadvantageous-to-AICN metric (iterations) for the sake of simplicity - to avoid dependency on the subproblem solvers for CN. Clearly, replacing metrics will improve the relative performance of AICN over CN. The extent of improvement is unknown yet, we need to rerun the experiments to see it.
> We are planning to do it before the next revision of the paper. In the meantime, we will change the introduction as it was written in abstract "our method has competitive performance“.
>
> ---
>
> >*In the end, I do think the global convergence of Newton's method is important and interesting for the NeurIPS community. The proposed method, AIC newton can potentially be a useful algorithm, but my main issue is that it has restrictive assumptions and no clear advantage over cubic newton.*
>
> There are two main advantages of AICN over Cubic Newton: **A)** simplicity, **B)** iteration cost.

---

> ### Author Response · Authors · 2022-08-02
> **Response to review (part 2)**
>
> >**Question:** *As the authors claim, one advantage of AIC Newton is that it is affine-invariant. This is indeed a natural consequence of replacing the ℓ2 norm by the local Hessian norm in the cubic term. But i'm not sure what is the real advantage here in terms of practical performance? What is the actual benefit of the method satisfying self-concordance?*
>
> **Response:** Advantage of affine-invariance is the inherent robustness to the choice of the basis it provides. A proper choice of the basis provides an extra layer of complexity for an algorithm. To support this, we provide a quote from section 5.1.2 of [3]: "*Newton’s method is affine invariant with respect to affine transformations of variables. Therefore, its actual region of quadratic convergence does not depend on a particular choice of the basis. It depends only on the local topological structure of the function* $f (x)$.”
>
> ---
>
> >**Question**: *Self-concordance is the property of the objective function. Our algorithm is designed to preserve it during the convergence process.
> In the numerical experiments, the authors say that the parameters Lf,L2 and α are all fine-tuned. How is this done exactly? In general, how would I pick Lf for AIC Newton?*
>
> We ran each algorithm multiple times with different parameters (more that 20 different values for each parameter for each problem) and reported the convergence of the run having the smallest function value.
>
> In practice, one can use some theoretical approximations (similarly to $L_2$) but we observed that the fine-tuned value of $L_f$ is much smaller and it leads to much faster convergence. Hence, one can do some tuning or try to use classical adaptive schemes.
>
> ---
>
> >**Question:** *In terms of per-iteration cost, does AIC Newton offer any advantage compared to cubic Newton of Nesterov? There has been many recent work that reduce the cost of the cubic subproblem [1] [2]. In light of these, how does of real running time of AIC newton compare with cubic Newton?*
>
> **Response:**
> In our experiments, we use matrix inversions via the built-in PyTorch function torch.linalg.solve. For Cubic Newton, we use line-search procedure from the original paper (Section 5.1); it requires multiple inversions or one eigenvalue decomposition in each iteration. On the other hand, AICN uses only one inversion per iteration.
>
> We are aware about articles [1] and [2], but we decided to not use them for the following reasons:
>
> - If we are using first-order methods to solve the subproblem, they needs $O(\varepsilon^{-1} \log (\varepsilon^{-1}))$ gradient steps to find the approximate solution of each subproblem. For Cubic Newton, a reasonable choice of the subsolver is Gradient Descent (GD) [1], which finds the minimum in $O(\varepsilon^{-1})$ iterations.
> For AICN, the subproblem is quadratic and one can utilize Conjugate Gradients (CG).
>
> - From our personal experience, when $x^k$ is far from the optimum, GD can solve the subproblem of Cubic Newton quickly. However, as it gets closer to a minimum, solving the subproblem gets harder - it requires a lot of GD steps. Also note that a good stepsize for GD changes for different $x^k$, hence one needs to tune it in each iteration.
> On the other hand, CG solves the AICN subproblems very quickly, and without any parameter tuning!
>
>
> ---
>
> **References:**
> - [1] Cubic regularization of Newton method and its global performance, Nesterov, Y. and Polyak, B.T., 2006, Mathematical Programming, 108(1), pp.177-205.
> - [2] Lower bounds for finding stationary points, Carmon, Y., Duchi, J.C., Hinder, O. and Sidford, A., 2020, Mathematical Programming, 184(1), pp.71-120.
> - [3] Lectures on Convex Optimization, Nesterov, Y., 2018

---

> ### Author Response · Authors · 2022-08-07
> **Reviewer 42nd: Did we manage to address your concerns?**
>
> Reviewer 42nd,
>
> Please could you let us know whether we managed to address your concerns?
>
> Thanks!!
>
> Authors

---

> > ### Comment · Reviewer_42nd · 2022-08-07
> > **Response to authors**
> >
> > I thank the authors for their detailed response and clarification on contributions. I have raised my score to 6.

---

### Official Review · Reviewer_xAoF · 2022-07-11

**Rating:** 6
**Confidence:** 5
**Soundness:** 3 good
**Presentation:** 2 fair
**Contribution:** 3 good

**Summary:**

In this submission, the authors proposed a new Newton's method called the Affine-Invariant Cubic Newton algorithm (AIC). This novel algorithm is the Newton's method with explicit new step size leading to the global convergence and local convergence. This AIC method can achieve a global convergence rate of $\mathcal{O}(1/k^2)$ which matches the state-of-art global convergence rate. On the other hand, this algorithm can also reach a local quadratic convergence rate which is the best-known local rate of second-order methods. One advantage of this algorithm is that the computation of step size is simple and doesn't need to solve the sub-problem. Another advantage is that the convergence results hold under the assumption that the objective function is affine-invariant, which is a good geometric property. The authors also conduct numerical experiments and the empirical results show that the performance of AIC is competitive with other state-of-art algorithms.

**Questions:**

In the checklist the authors said that they discussed the limitations of this proposed algorithms. Could you please summarize the limitations in one final conclusions section after the numerical experiments section?

**Limitations:**

The authors didn't include limitations in the paper. They should add one conclusion section summarizing the paper and discussing any potential limitations and future work directions after the numerical experiments section.

**Strengths And Weaknesses:**

This paper has the following strengths in terms of the originality, significance, quality and clarity:

Originality and significance: this paper provides the first variant of Newton's method that could achieve state-of-art convergence rates in both terms of the global and local convergence for the objective function satisfying the affine-invariant property. And the computation of the step size is not complicated. This is the main and independent contribution of this paper.

Quality and clarity: the authors provides detailed and clear explanations in both the theoretical analysis section and the numerical experiments section. The theoretical proof in the appendix is correct. The empirical results show that the proposed novel algorithm is competitive against other state-of-art methods. The paper is also organized with clear and clean structure and the language is well-organized so that the readers can easily understand the content of this submission.

However, despite the above advantages, the paper has the following weaknesses:

The main issue is that the parameters in definition 2, 3 and 4 are expressed a bit unclear. Is this $L_f$ the same for all assumptions 2,3,4? Or are they different? Then, the authors should use different notations to express them. Also in the Theorem 2 and 3, the authors should mention the exact parameter $L_f$ for the assumptions of weakly-strongly self-concordance and semi-strongly self-concordance.

Another issue is that the authors presented the definitions 2, 3, 4 explicitly and showed the relationship of them. Are these assumptions necessary for the theoretical analysis of the AIC algorithm? Can we obtain the similar results with only the assumption of self-concordance? If these global and local convergence rates also hold for the only assumption of standard self-concordance, then there is no need to create these new assumptions and this submission will be much more clear and simple.

Also, for all these variants of Newton's method, the authors should mention that the computational expense per iteration is $\mathcal{O}(d^3)$ where $d$ is the dimension of the problem. This is because we need to compute the inverse of the Hessian matrix to obtain the step size at each iteration. And this expense is very high, which is the main limitation of the Newton's method. In term of the computational cost per iteration, the proposed new AIC algorithm is the same as all the previous variants of Newton's method.

The last minor issue is that in Algorithm 1, what's $L_H$? Is this a typo and it should be $L_f$?

---

> ### Author Response · Authors · 2022-08-02
> **Response to review**
>
> >*The main issue is that the parameters in definition 2, 3 and 4 are expressed a bit unclear. Is this  Lf the same for all assumptions 2,3,4? Or are they different? Then, the authors should use different notations to express them. Also in the Theorem 2 and 3, the authors should mention the exact parameter Lf for the assumptions of weakly-strongly self-concordance and semi-strongly self-concordance.*
>
> **Regarding definitions:** Thank you for pointing out this confusion. We will simplify this part by removing Definition 2 and use only Definition 3 in all theorems. We will also change the notation of $L_f$ to distinguish those constants.
>
> ---
>
> >*Another issue is that the authors presented the definitions 2, 3, 4 explicitly and showed the relationship of them. Are these assumptions necessary for the theoretical analysis of the AIC algorithm? Can we obtain the similar results with only the assumption of self-concordance? If these global and local convergence rates also hold for the only assumption of standard self-concordance, then there is no need to create these new assumptions and this submission will be much more clear and simple.*
>
> **Regarding self-concordance:** Unfortunately, standard self-concordance is not enough to show fast global convergence for AICN. The reasons are as follows.
>
> For Euclidean norm, conditions
> - (A) $\Vert \nabla^2 f(y) - \nabla^2 f(x) \Vert \leq L_2 \Vert y-x \Vert$
> - (B) $ D^3 f(x)[y][h]^2\leq L_2 \Vert y \Vert \ \Vert h \Vert^2$
>
> are equivalent if $f(x)$ have continuous third-derivative. We can get (A) from (B) by integration technique:
> $$\left\langle (\nabla^2 f(y)-\nabla^2 f(x)) h,h \right\rangle \leq \int\limits_0^1 D^3 f(x+\tau(y-x))[y-x][h]^2 d\tau\leq L_2 \Vert y-x \Vert \ \Vert h \Vert ^2.$$
> You can see that the right-hand side of (B) does not depend on the point $x$ of the third-derivative. But for self-concordance the right-hand side will be $\Vert y \Vert_x$, so we can not simplify this integral in the same way. For more details, you can check Subsection 5.1.2. of [1].
>
> ---
>
> >*Also, for all these variants of Newton's method, the authors should mention that the computational expense per iteration is $O(d^3)$ where d is the dimension of the problem. This is because we need to compute the inverse of the Hessian matrix to obtain the step size at each iteration. And this expense is very high, which is the main limitation of the Newton's method. In term of the computational cost per iteration, the proposed new AIC algorithm is the same as all the previous variants of Newton's method.*
>
> **Regarding iteration cost:** As AICN is a second-order method, we compare it to other second-order methods (Cubic Newton, Damped Newton). Yes, all of them need $O(d^3)$ operations to invert matrices in a classical way.
> However, note that Cubic Newton has a subproblem that requires additional line search and inverting the Hessian matrix $O(\log \varepsilon^{-1})$ times in each iteration. Hence, the iteration cost of Cubic Newton is $O(d^3\log \varepsilon^{-1})$ while for AICN it is just $O(d^3)$. For more details, you can check Subsection 4.1.4 of [1] or the original paper about Cubic Newton [2]. We will point this out in the paper.
>
> ---
> >*The last minor issue is that in Algorithm 1, what's LH? Is this a typo and it should be Lf?*
>
> Good catch. Yes, it is a typo; we corrected it.
>
> ---
> >*The authors didn't include limitations in the paper. They should add one conclusion section summarizing the paper and discussing any potential limitations and future work directions after the numerical experiments section.*
>
> **Limitations:** Thanks for the advice, we will mention those explicitly. In practice, AICN suffers from same limitations as others Newton-like methods: their numerical stability depends on the conditioning of the Hessian. They are memory and computationally consuming for high-dimensional problems.
>
> **Future work:** There are many natural extensions of AICN. We are currently working on the following: how to avoid working with full Hessian (how to allow for stochasticity), how to accelerate the method, extensions to general nonconvex functions…
>
> ---
>
> **References:**
> - [1] Lectures on Convex Optimization, Nesterov, Y., 2018
> - [2] Cubic regularization of Newton method and its global performance, Nesterov, Y. and Polyak, B.T., 2006, Mathematical Programming, 108(1), pp.177-205.

---

> > ### Author Response · Authors · 2022-08-07
> > **Reviewer xAoF: Did we address your concerns?**
> >
> > Reviewer xAoF,
> >
> > Please could you let us know whether we managed to address your concerns?
> >
> > Thanks!!
> >
> > Authors

---

> > > ### Comment · Reviewer_xAoF · 2022-08-07
> > > **Response to the authors**
> > >
> > > I appreciated the authors for their detailed response and clarification on each comments. I remained my point of view. Thanks.

---

### Author Response · Authors · 2022-08-02
**Summary of reviews**

We would like to thank all reviewers for their reviews. We would like to highlight following points:

**Motivation + writing:**
- “The authors provides detailed and clear explanations in both the theoretical analysis section and the numerical experiments section.” (reviewer xAoF)
- “The paper is also organized with clear and clean structure and the language is well-organized so that the readers can easily understand the content of this submission.” (reviewer xAoF)
- “This article is well-written and offers an overview of potential problems for Newton's method and how these problems were previously addressed.” (reviewer 42nd)
- “The authors motivate their algorithm, which they call AIC Newton, quite naturally and the idea behind the proofs are also nicely communicated.” (reviewer 42nd)
- “The paper is very well-written and easy to follow.” (reviewer dYEq)


**Theoretical contributions:**
- “The contributions are elegant and significant.” (reviewer dYEq)
- “In the end, under certain assumptions, the proposed algorithm is easy to implement and has strong local and global guarantees.” (reviewer 42nd)
- “The theoretical proof in the appendix is correct.” (reviewer xAoF)


**Experiments:**
- “The empirical results show that the proposed novel algorithm is competitive against other state-of-art methods.” (reviewer xAoF)
- “There are compelling numerical experiment.” (reviewer dYEq)

---

Reviewers xAoF and 42nd pointed out some minor concerns and areas where we can further improve writing. We are grateful for those, we are improving the paper in the following regards:
- We are clarifying the difference of AICN compared to Cubic Newton method (iteration cost of Cubic Newton, plots,...).
- We are simplifying the self-concordance comparison (cleaning notation + removing extra definition (Def. 2)).
- We are adding the limitations of Newton-like methods to conclusion.

We believe that we addressed all concerns raised by the reviewers. Reviewer 42nd said “However, there are a few issues that prevent me from giving this article a higher score.” In this regard, we would like to kindly ask reviewers to increase their score accordingly.

If you have any more questions, we are happy to engage in the further discussion.

---

### Meta-Review · Area_Chair_TBZJ · 2022-08-21

**Recommendation:** Accept
**Confidence:** Certain

**Metareview:**

There is general agreement that this paper should be accepted.

**Award:**

No

---

### Decision · Program_Chairs · 2022-09-14

Accept